

# Weekly Green Tide Mapping in the Yellow Sea with Deep Learning: Integrating Optical and SAR Ocean Imagery

Le Gao[1], Yuan Guo[1], Xiaofeng Li[1]

[1]Key Laboratory of Ocean Circulation and Waves, Institute of Oceanography, Chinese Academy of Sciences, Qingdao, 266071, China

*Correspondence to*: Xiaofeng Li (lixf@qdio.ac.cn)

**Abstract.** Since 2008, the Yellow Sea has experienced a world's largest-scale marine disasters, known as the green tide, marked by the rapid proliferation and accumulation of large floating algae. Leveraging advanced AI models, namely AlgaeNet and GANet, this study comprehensively extracted and analyzed green tide occurrences using optical Moderate Resolution Imaging Spectroradiometer (MODIS) images and microwave Sentinel-1 Synthetic Aperture Radar (SAR) images. Most importantly, this study presents a continuous and seamless weekly average green tide coverage dataset with the resolution of 500 m, by integrating high precise daily optical and SAR data during each week during the green tide breakout. The uncertainty assessment of this weekly product shows it is completely consistent with the overall direct average of the daily product ($R^2$=1 and RMSE=0). Additionally, the individual case verification in 2019 also shows that the weekly product conforms to the life pattern of green tide outbreaks and exhibits parabolic curve-like characteristics, with an low uncertainty ($R^2$=0.89 and RMSE=275 $km^2$).This weekly dataset offers reliable long-term data spanning 15 years, facilitating research in forecasting, climate change analysis, numerical simulation and disaster prevention planning in the Yellow Sea. The dataset is accessible through the Oceanographic Data Center, Chinese Academy of Sciences (CASODC), along with comprehensive reuse instructions provided at http://dx.doi.org/10.12157/IOCAS.20240410.002 (Gao et al., 2024).

## 1 Introduction

Situated between China and Korea, the Yellow Sea (illustrated in Fig. 1) is a marginal sea with abundant biodiversity. The Yellow Sea green tide presents a formidable ecological challenge within this maritime expanse. Comprising primarily of large floating algae, notably *Enteromorpha*, these algae proliferate and aggregate under particular environmental conditions, culminating in marine ecological disasters. The Yellow Sea green tide showcases distinctive seasonal and spatiotemporal distribution patterns. Since 2008, they have occurred annually from early May to late August, traversing from the Subei Shoal in the western Yellow Sea to the Shandong Peninsula in the northern Yellow Sea (Fig. 1). Throughout this migration, green algae undergo rapid proliferation and aggregation, forming the world's most extensive green algal belts (Liu et al., 2013; Wang et al., 2015; Valiela et al., 2018). Changes in the drift patterns and strength of these green tide blooms could



significantly affect the offshore fishery resources, ecological environment, and tourism industry of the Yellow Sea (Cao et al.,
2020). Consequently, monitoring and analyzing the Yellow Sea green tide remain imperative and pressing tasks.

Due to its extensive coverage and rapid revisit capabilities, satellite remote sensing technology has emerged as the predominant method for spatiotemporal monitoring of large floating algae. Previous research has leveraged optical and synthetic aperture radar (SAR) satellite imagery to track the entire life cycle of green tides (see Fig. 1a-b). Optical satellite sensors, typified by the Moderate Resolution Imaging Spectroradiometer (MODIS) onboard the Terra and Aqua satellites,
the Multi-Spectral Scanner (MSS),Thematic Mapper (TM),Enhanced Thematic Mapper Plus(ETM+), Operational Land Imager (OLI) and Thermal Infrared Sensor(TIRS) onboard Landsat (A series of Earth-observing satellite missions since 1972), Visible Infrared Imaging Radiometer Suite (VIIRS) onboard the Suomi National Polar-Orbiting Partnership (Suomi NPP) spacecraft, and Geostationary Ocean Color Imager (GOCI) onboard the Communication, Ocean and Meteorological Satellite (COMS) have become the primary data source for extracting green tide information (Xing et al., 2016 and 2018).
As illustrated in Fig. 1a, green algae manifests as a distinct red strip in the false-color image synthesized from near-infrared,red, and green bands, presenting a stark contrast with the surrounding seawater. However, owing to the spectral reflectance similarity between green algae and terrestrial vegetation, numerous biological indices (such as Normalized Difference Vegetation Index (NDVI), Floating Algae Index (FAI), and Alternative Floating Algae Index (AFAI) have been proposed and utilized for green algae extraction (Hu, 2009; Son et al., 2012; Fang et al., 2018). Yet, due to the coarse
resolution and mixed pixels of optical satellite images, only relatively large green algae strips can be identified, resulting in both overestimation and underestimation issues in optical imagery (Cui et al., 2018 and 2020). Moreover, despite the daily transit of optical satellites over the Yellow Sea area, the continuous observation of green tide information is impeded by the presence of clouds and rain.

SAR presents an alternative effective tool for monitoring green tides, exemplified by Sentinel-1 and Gaofen-3. Unaffected
by clouds and rain, SAR sensors operates under all-weather conditions and is unaffected by clouds and rain (Qi et al., 2022b). SAR emits radar pulses using several polarization modes and retrieves backscattering signals from the sea surface, known as normalized radar cross section (NRCS). NRCS is commonly affected by Bragg waves generated by winds and currents. As green algae float on the sea surface and mimic solid objects, they produce significant volume scattering or double- triple-bouncing in the incoming radar signal due to their solid structure. Numerous methods have been proposed to extract green
tide information from radar NRCS data under different polarization modes, including threshold-based automatic approaches and empirical human threshold methods (Yu et al., 2020; Ma et al., 2022). Essentially, these methods classify green algae based on the contrast between algae and seawater. With higher resolution, narrower observation swaths, and long revisit cycles (e.g., Sentinel-1's single satellite with a 12-day cycle, and a constellation of two satellites enabling global image capture every 6 days), microwave SAR imagery can capture even the smallest patches of green algae (Fig. 1b). However,
their coverage of the entire Yellow Sea remains incomplete, as depicted in Fig. 1b. Green algae strips exhibit strong reflectivity, appearing as bright white patches on SAR images, while seawater seems black. As microwaves cannot penetrate





seawater, SAR sensors only capture green algae completely floating on the sea surface, thus omitting information about green algae below the seawater surface (Gao et al., 2022).

Therefore, integrating the green algae extraction results from both optical MODIS and microwave SAR systems not only

enhances the effective number of daily observations throughout the algae's life cycle but also addresses the limitations of each system. While optical MODIS typically observes large green algae strips due to their coarse resolution, SAR primarily detects green algae on the sea surface. Combining these datasets generates a weekly product, providing a comprehensive representation of continuous green tide changes.



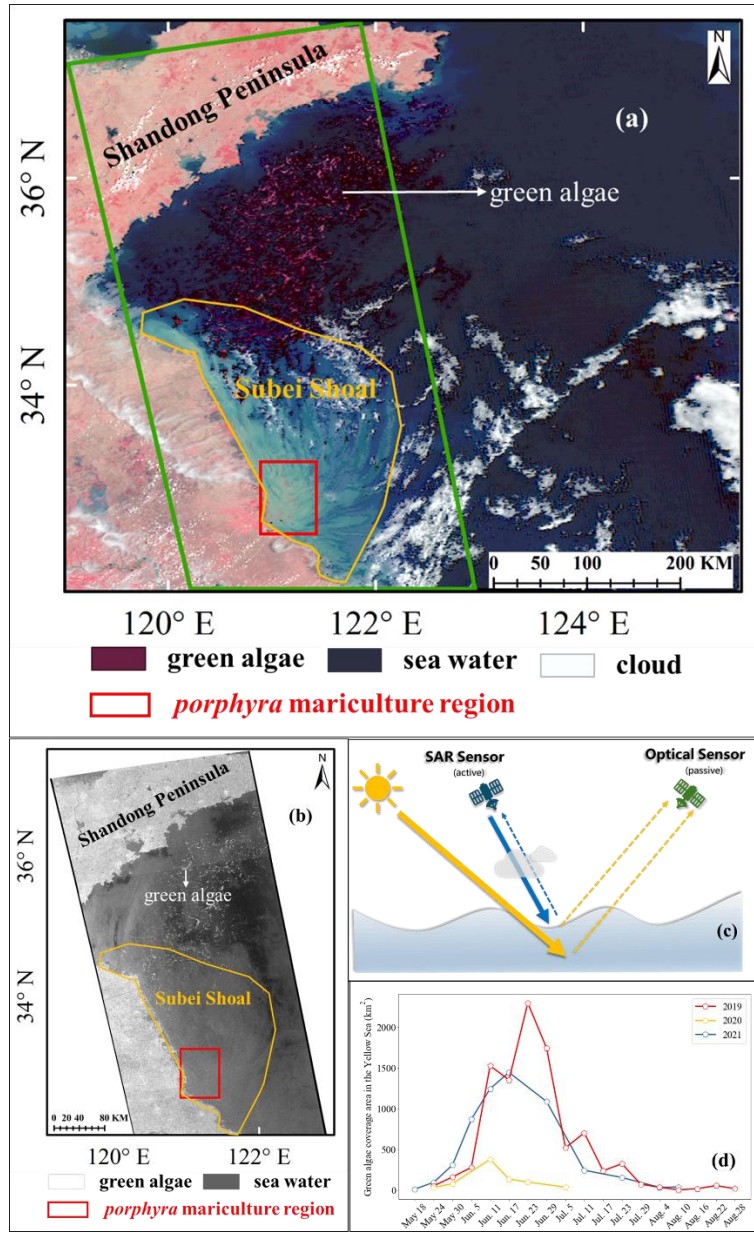

**Figure 1. The occurrence scale of green tides in the Yellow Sea.** (a) False-color image synthesized from near-infrared, red, and green bands; (b) Sentinel-1 SAR image captured within the Yellow Sea and corresponds to the location indicated by the green box in Figure (a); (c) Schematic diagram depicting the optical and SAR imaging processes; (d) Randomly selected daily green tide coverages in 2019-2021, revealing a distinct "parabola-like" pattern.

In recent years, propelled by the rapid advancements in artificial intelligence (AI) technology (Jordan et al., 2015; LeCun et al., 2015; Li et al., 2020; Dong et al., 2022; Li et al., 2022; Chen et al., 2023;Wang and Li, 2024), several green tide





extraction algorithms based on deep learning have emerged. Notably, models such as the AlageNet (Gao et al., 2022) and GANet (Guo et al., 2022) leverage image texture enhancement mechanisms and attention mechanisms, effectively addressing the challenge of algae-water imbalance for the optical MODIS and Sentinel-1 SAR imagery. These models boast

superior detection accuracy and generalization capabilities compared to some state-of-the-art models, e.g., classic U-Net, VGG-16, Random Forest (RF) and normalized difference vegetation index (NDVI). They eliminate the need for fixed threshold selection, ensuring consistent green algae detection across diverse imaging conditions. However, the efficacy of AI models hinges on the availability of abundant representative green algae training samples across various environmental scenarios, necessitating labor-intensive and time-consuming manual labeling efforts. Consequently, manual labeling and

sharing representative sample labels have posed persistent challenges. Moreover, despite the plethora of green algae extraction algorithms proposed in previous studies, the time series of historical green tide coverage datasets, dating back to the inception of green tide records, have yet to be made publicly available and shared (Hu et al., 2019; Hu et al., 2023; Cao et al., 2023). These datasets serve as the foundation of green tide research and provide essential data for the mutual comparison and verification of various extraction algorithms.

Recently, through in-depth research efforts, there has been some preliminary understanding of the mechanisms underlying green tide outbreaks (Zhang et al., 2019; Feng et al., 2020; Cao et al., 2023). The genesis of green tide is commonly attributed to Porphyra mariculture in the Subei Shoal, as depicted in Fig. 1a-b (Xing et al., 2019). It is believed that green algae seed spores detach from Porphyra mariculture rafts and disperse into the seawater, proliferating rapidly under favorable environmental conditions, such as suitable sea surface temperature (SST) and nutrient richness, ultimately leading to large-

scale outbreaks (Li et al., 2015; Li et al., 2016). The outbreak of green tide is influenced by a combination of environmental and anthropogenic factors, with significant inter-annual variations in outbreak magnitude (Guo et al., 2016), including the conspicuous "parabola-like" pattern, illustrated in Fig. 1d. However, due to the lack of a comprehensive and scientific understanding of the green tide bloom mechanism, the root cause of the "jumping" changes in green algae coverage area remains elusive. Particularly when various factors are interdependent and mutually influencing each other (Jin et al., 2018;

Li et al., 2021a and 2021b), the absence of collaborative joint analysis impedes accurate elucidation of the critical response mechanism and interactions between environmental factors and green tide outbreaks. Hence, it is imperative to investigate the interplay between green tide occurrences and the environment by examining the correlation between environmental factors and outbreaks of green algae.

The objectives of this article are twofold, as shown in Fig. 2:

1) Develop a weekly green algae coverage dataset by integrating optical and SAR data. This dataset addresses limitations such as missing small strips in optical images due to low resolution and the inability of SAR to observe green algae not completely floating on the sea surface. It also aims to overcome the "discontinuous" characteristics of previous daily green algae coverage datasets.



2) Analyze the relationship between green tide coverage and various environmental factors influencing its occurrence, leveraging the spatiotemporal characteristics of green tide outbreaks. This analysis aims to identify the factors contributing to variations in outbreak scale and assess the impact of green tide activities on the ecological environment.

This study manually annotates optical and SAR images across diverse environmental conditions to achieve these goals and creates and shares representative green tide sample datasets tailored to multi-environment scenarios. These sets will be training data for developing other AI-based green algae extraction models. The AlageNet and GANet models will also be refined and retrained for detecting green tides in optical MODIS and microwave Sentinel-1 SAR images, respectively. This continued training step facilitates the creation of daily green tide coverage datasets. The accuracy of daily green tide detection will be evaluated by comparing the results with the manually annotated sample dataset and fully annotated images. Furthermore, differences in green tide detection under the two observation modes of optical and microwave will be analyzed, followed by the validation of uncertainty in the weekly datasets using skillful assessment strategies.

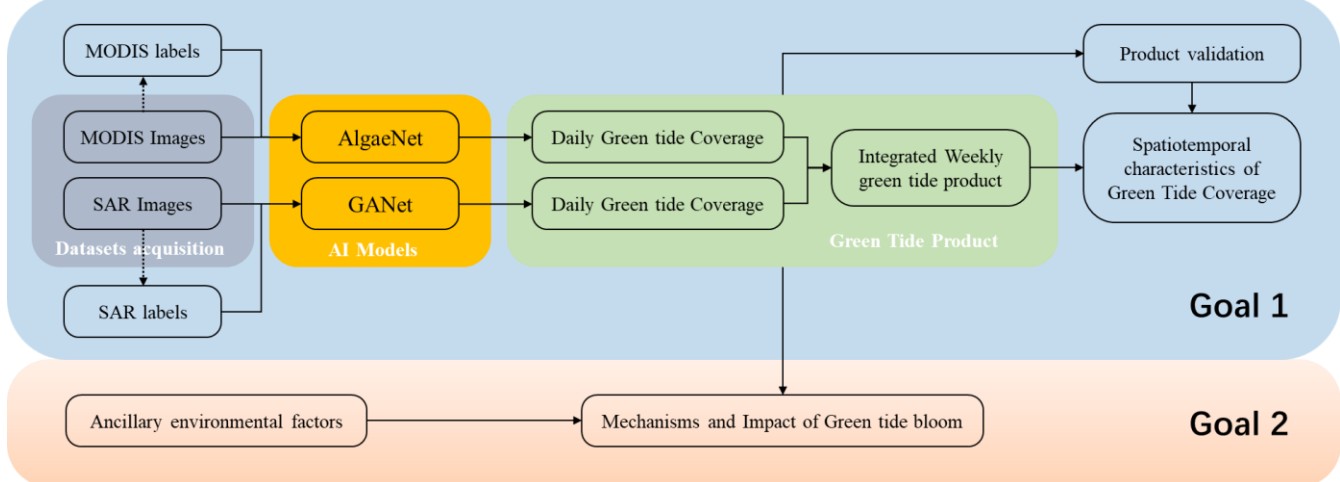

**Figure 2. Overall flow chart of green tide coverage products generation.**

## 2 Data and Methods

### 2.1 Datasets acquisition.

The optical MODIS and Microwave SAR images and the ancillary environmental factors affecting green tide changes in the Yellow Sea are acquired. These datasets are utilized to construct our green tide coverage time series and to analyze the relationship between green tide bloom and the environment.



### 2.1.1 Optical MODIS imagry

MODIS, comprising Terra and Aqua satellites launched in 1999 and 2002, respectively, has maintained stable operations,
enabling daily observations covering the entire Yellow Sea. Since 2008, the Yellow Sea green tide began to be recorded by satellites. This study utilizes optical MODIS surface reflectance products, specifically MYD09GA and MOD09GA, which offer data for Bands 1-7 in a daily gridded L2G product. Different combinations of these bands produce distinct false-color images. Notably, within the electromagnetic spectrum, near-infrared (Band 2) bands exhibit prominent peaks compared to red (Band 1) and green (Band 4) bands (Qi et al., 2017). Consequently, in false-color images generated by these bands (Fig.
3a), the green tide appears red, providing enhanced contrast with the surrounding seawater. This approach offers notable advantages over true-color images created using the red (Band 1), green (Band 4), and blue (Band 3) bands depicted in Fig. 3. From 2008 to 2022, we collected a total of 577 daily optical images, comprising 258 from Aqua and 319 from Terra, with a resolution of 500 meters in the Yellow Sea. These MODIS images are geometrically and radiometrically corrected. Since optical sensors possess certain underwater detection capabilities, MODIS can effectively detect green algae on the water
surface and submerged portions.

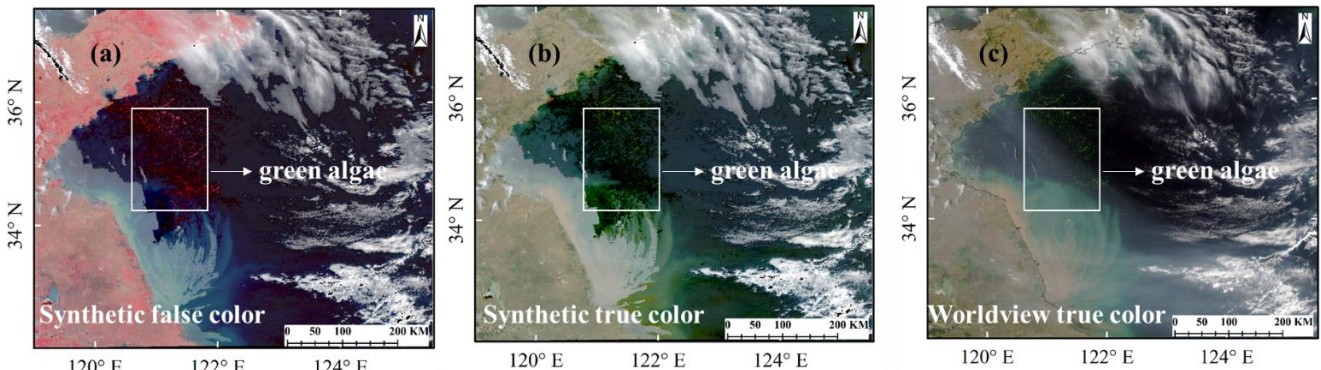

**Figure 3. Synthetic images of near-infrared, infrared, and green bands (date: June 23, 2021).**

### 145 2.1.2 Microwave SAR imagery

The Sentinel-1 satellite, composed of SAR satellites A and B launched in 2014 and 2016 respectively, operates on a 12-day repeat cycle, providing global coverage every 6 days through a satellite constellation. It began retrieving green algae data in the Yellow Sea from 2015 onwards. As a result, the daily green tide coverage dataset released spans from 2015 to 2022, with a time resolution of a 6-day cycle based on Sentinel-1 SAR satellite images. This study utilizes microwave Sentinel-1 SAR
satellite imagery. We collected 216 Sentinel-1 Level-1 ground range detected high-resolution (GRDH) images with VV and VH polarizations spanning the Yellow Sea from 2015 to 2022. These SAR images are acquired in interferometric wide (IW) mode, featuring a 250-kilometer swath and 10-meter initial resolution. At this high resolution, Sentinel-1 images can detect



small algae patches. However, due to the rapid absorption of microwave signals in the water, the SAR sensor can only capture reflected signals from green algae entirely floating on the sea surface, resulting in the observed green tide appearing

as a bright white strip in SAR imagery. The dataset comprising SAR images covering the Yellow Sea is extensive, with individual images exceeding 1 gigabit in size. To enhance the processing efficiency of satellite images containing green tides, we resampled the original 10-meter resolution to 30 meters.

### 2.1.3 Ancillary environmental factor data

The environmental data utilized in this study are sourced from various atmosphere-ocean models, as referenced in works

such as Li et al. 2022a and 2022b. These data stem from four primary model assimilation/reanalysis sources, detailed in Table 1: the HYbrid Coordinate Ocean Model (HYCOM) provided sea surface temperature (SST), sea surface salinity (SSS), and sea surface circulation (SSC) at a resolution of 0.08°; The the fifth-generation atmospheric reanalysis data (ERA5) of the European Centre for Medium-Range Weather Forecasts (ECMWF) offers sea surface wind (SSW) and precipitation (PRCP) at a resolution of 0.25°; The Copernicus Marine Environment Monitoring Service (CMEMS) provided nutrients and

dissolved oxygen (O2) at the same resolution of 0.25°. Additionally, the Goddard Space Flight Center (GSFC) contributes solar radiation (SIR) at a resolution of 0.25°. These sources amalgamate model data with global observations, yielding a comprehensive and consistent dataset that effectively simulates the coastal waters of the China Sea (Xu et al., 2011; Wang et al., 2020). To ensure consistency across different data sources, we resampled all aforementioned environmental elements to a resolution of 500 meters, aligning with the resolution of MODIS.

**Table 1. Major environmental elements**

| Data source | Potential environmental factors | Original Resolution | Download address | Derived data |
|---|---|---|---|---|
| HYCOM | **SST**: sea surface temperature | 0.08° | https://developers.google.com/earth-engine/datasets/catalog/HYCOM_sea_temp_salinity | Derived gradient field |
| | **SSS**: sea surface salinity | | https://developers.google.com/earth-engine/datasets/catalog/HYCOM_sea_temp_salinity | |
| | **SSC**:sea surface circulation | | https://developers.google.com/earth-engine/datasets/catalog/HYCOM_sea_water_velocity | |
| ECMWF/ERA5 | **SSW**: sea surface wind | 0.25° | https://developers.google.com/earth-engine/datasets/catalog/ECMWF_ERA5_DAILY | |
| | **PRCP**: precipitation | | | |
| CMEMS | **Nutrients** (Nitrate, Phosphate) | | https://data.marine.copernicus.eu/product/GLOBAL_ANALYSIS_FORECAST_BIO_001_028/description | Derived gradient field |
| | O2: Dissolved oxygen | | | |
| GSFC/ NASA Earth Observations | **SIR**:Solar radiation | | https://oceandata.sci.gsfc.nasa.gov/ | |

## 2.2 Data annotation

To develop an AI-based green algae extraction model, precise training samples are imperative as ground truth for model training. We conduct expert-level visual interpretation and manual annotation of optical MODIS and Sentinel-1 SAR images to achieve this.

### 2.2.1 Optical image annotations

To establish precise ground-truth labels for optical images, we rely on a carefully selected set of 48 false-color images derived from the near-infrared, red, and green bands as the basis for visual interpretation and labeling of green tides. In regions encompassing oceanic deep water areas devoid of clouds, offshore shallow water regions without clouds, thin cloud areas, dense and sparse algae strip regions, and cloud edge zones, algae-containing pixels are manually identified on optical images using Labelme software (Russell et al., 2008). A total of 5,296 pairs of samples are labeled as training sets and 662 pairs as testing sets, with each labeled sample standardized to 128×128 pixels.

### 2.2.2 SAR image annotations.

To enhance the feature learning capabilities of the green tide detection model across diverse environmental contexts, including dense green algae strip areas, sparse strip areas, oceanic deep water regions, and offshore shallow water areas, we utilize a resolution of 30 meters while maintaining a size of 256×256 pixels for the training samples. Maintaining the size of the training sample while transitioning to a 30-meter resolution enables a broader representation of green tide characteristics, owing to the increased spatial scale compared to the original 10-meter resolution. For instance, a training sample featuring green algae features across both deep and shallow water regions allows the detection model to learn from these varied features concurrently. Conversely, training samples at 10-meter resolution often exhibit uniform green tide characteristics, limiting the deep-learning model's learning capacity. Similarly, we manually annotate algae-containing pixels on a meticulously selected set of 22 SAR images using Labelme software. This annotation process results in a total of 4,535 pairs of samples, comprising 4,268 pairs for training and 267 pairs for testing, facilitating comprehensive model training and evaluation across a spectrum of environmental scenarios.

## 2.3 Deep-learning models

Recently, we introduced two AI-based algorithms, AlgaeNet (Gao et al., 2022) and GANet (Guo et al., 2022), designed for the rapid and precise extraction of green tide coverages from optical and SAR imagery (see Fig. 4). Both models demonstrate significant scalability and can be readily applied to various satellite images, including optical MODIS/GOCI/Landsat and microwave Sentinel-1/Gaofen-3/RadarSat.



### 2.3.1 AlgaeNet model

AlgaeNet model addresses both physical-ware input and algae-water imbalance in training samples, mitigating potential biases inherent in traditional threshold-based segmentation methods. It outperforms other models, such as random forest and VGG16, in accuracy for optical MODIS imagery and achieves higher recall and precision than optical index methods like NDVI/FAI/EVI (Gao et al., 2022). Figure 4a shows the AlgaeNet model's system diagram based on the U-Net framework, including input, encoder, decoder, and prediction modules. Unlike the original AlageNet model, this study further modified the input module. We used the unique physical multichannel combination of all bands of MODIS surface reflectance products as input, and the improved AlgaeNet model can perform green algae detection in the optical image. All pixels of the entire area are divided into two categories: seawater pixels and algae pixels.

We employed the AlgaeNet model to extract daily green tide coverages from optical MODIS images. Unlike previous methods that solely utilized true-color bands (Gao et al., 2022), our approach incorporates MYD09GA and MOD09GA channels (MODIS surface reflectance products), enabling more precise detection in areas affected by thin clouds, shallow waters, and turbid seawater. We further enhanced extraction accuracy by retraining the model with a new annotation dataset (see Table 2), achieving a mean intersection over union (mIOU) of 67.51%. However, MODIS images are susceptible to cloud edge effects (see Fig. 5), leading to the misidentification of some of the green algae pixels. To address this challenge, we implemented a filtering strategy, eliminating green algae patches smaller than 1.10 km2 to mitigate misidentifications at broken cloud boundaries while retaining algae information in cloud-free and thin cloud areas.

### 2.3.2 GANet model

GANet model, incorporating attention mechanisms, sample imbalance loss functions, and texture enhancement mechanisms, surpasses alternative algorithms (Guo et al., 2022). Figure 4b illustrates the system diagram of the GANet model, built upon the U-Net framework. It comprises the input, encoder, decoder, and prediction modules. Distinguished from the original GANet model, two significant enhancements have been introduced to the texture enhancement mechanisms utilizing 30 m resolution SAR images, replacing the prior 10 m resolution images. Primarily, the input module now integrates VV-/VH-polarized NRCS data and textural feature maps derived from the SAR dataset, specifically employing the gray-level co-occurrence matrix (GLCM). GLCM features are also extracted into pooling layers, generating feature maps within the encoder and decoder modules. The resulting multiscale GLCM features are then concatenated with feature tensors generated by the convolutional layers. Moreover, when feeding image slices into the AlgaeNet model, we apply random brightness enhancements to the image slices to improve the model's adaptability to different sea conditions. These enhancements empower the GANet model to effectively discern green algae in SAR images, thereby partitioning all pixels across the entire area into seawater and algae classifications.

We employed the GANet model to extract daily green tides from Sentinel-1 SAR images and augmented the model's accuracy by integrating additional training samples from shallow nearshore waters. Furthermore, adopting 30 m resolution



samples with a size of 256   256 pixels expanded the model's feature learning ability compared to the previous 10 m resolution. These enhancements have raised the model's performance to 85.41% (see Table 2), although this appears similar to the original model's accuracy of 86.31%, it's important to note that the comparison is affected by inconsistent label sample

sizes. Additionally, these enhancements have significantly improved processing performance and efficiency across the entire imagery of the Yellow Sea region, transcending the limitations of a limited number of labeled samples.

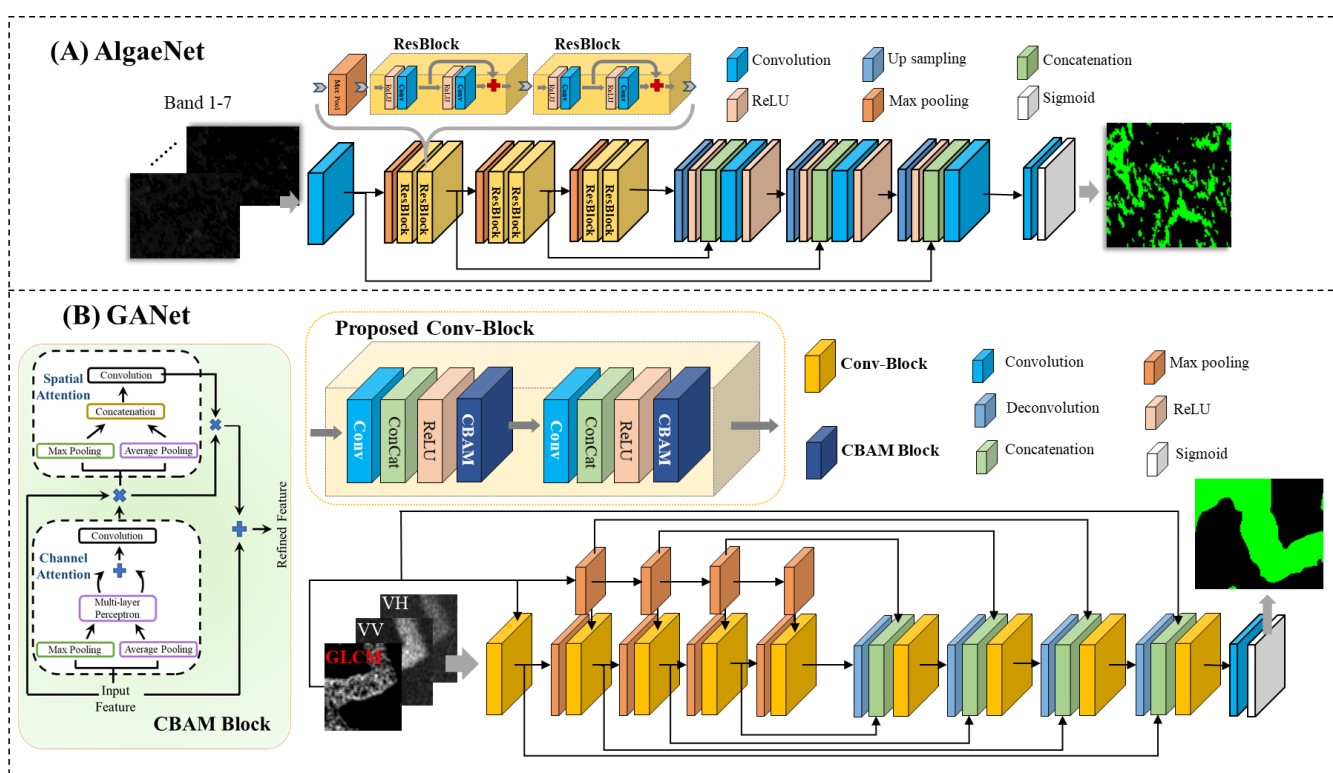

**Figure 4. Green tide detection network.** (a) Proposed AlgaeNet model based on the basic UNet framework; (b) Proposed GANet model

based on the basic U-Net framework.

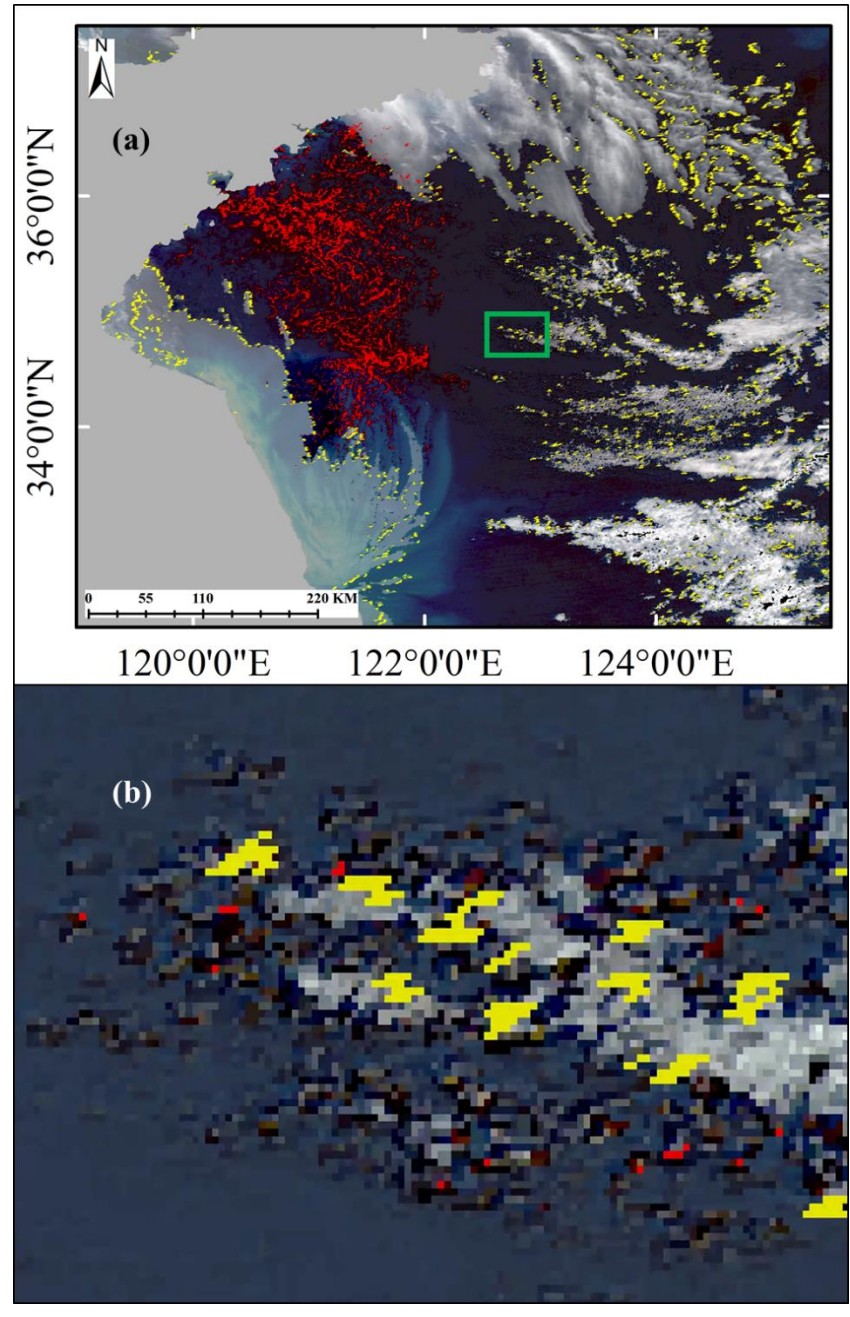

**Figure 5. Size statistics of misidentified patches at the cloud edge (date: June 23, 2021).** (a) Randomly selected MODIS optical image, where red patches represent green algae pixels and yellow dots indicate cloud shadows derived from MODIS product; (b) an enlarged view of the white square part in (a), with the red pixels representing algae pixels that were mistakenly identified due to the presence of cloud edges. Initially, they were supposed to be eliminated using yellow shadows. However, since the two do not coincide, the misidentification cannot be rectified.



**Table 2.** Accuracy of daily green tide detection model based on the testing set (%)

| Type | Accuracy | Precision | Recall | F1-score | mIOU | Number of testing samples | |
|---|---|---|---|---|---|---|---|
| Optical | **99.26** | **87.67** | **73.88** | **80.60** | **67.51** | 662 | This study |
| | 97.03 | 75.36 | 57.73 | 65.38 | 48.57 | 316 | Gao et al., 2022 |
| SAR | **99.82** | **94.69** | **89.71** | **92.13** | **85.41** | 267 | This study |
| | 98.36 | 93.29 | 92.03 | 92.65 | 86.31 | 2124 | Guo et al., 2022 |

## 2.4 Integrating daily optical and SAR products

The direct outputs from the AlgaeNet and GANet models are daily green tide coverage and distribution, with a spatial resolution of 500 m from MODIS images and 30 m from Sentinel-1 SAR images, respectively. While MODIS images provide comprehensive coverage of the entire Yellow Sea during the green tide period, cloud and rain interference limit effective green tide observations to 2-4 times per week under cloud-free or thin cloud conditions. Consequently, daily green tide coverages derived from optical images may still exhibit several missing frames on certain days. On the other hand, the Sentinel-1 satellite operates with a time resolution of a 6-day cycle. However, the effective observation range of green tides by Sentinel-1 SAR sensors is primarily limited to most regions of the Yellow Sea (Fig. 1b) but not the entire Yellow Sea. This discontinuous green tide coverage challenges practical applications such as green tide forecasting. Therefore, a fusion of these two types of daily products is necessary to produce continuous and seamless green tide data products.

Previous studies, including Li et al. (2021b), have proposed various methods to integrate green tide datasets derived from optical and SAR images, aiming to enhance compatibility and extend the temporal sequences of daily green tide observations. Figure 6 illustrates simultaneously observed optical and SAR images alongside their corresponding green tide coverages (Fig. 6a-b). The optical green tide coverage pattern aligns seamlessly with that captured by SAR imagery (Fig. 6c). However, while optical sensors can detect algae strips both on the sea surface and beneath a certain water depth, SAR sensors only capture signals from algae strips entirely floating on the surface (as illustrated in Fig. 1c). Consequently, the boundaries of algae strips detected by optical sensors tend to appear wider for larger green algae strips. Additionally, due to the relatively coarse resolution of the optical MODIS sensor, very small green algae patches may be missed (Fig. 6d). For instance, Li et al. (2021b) proposed a method to standardize algae detection results from high-resolution images to a coarser resolution. However, this approach is primarily suitable for larger green algae strips detectable by both sensors and may not adequately address tiny green algae patches overlooked in optical images. This forced standardization strategy inadvertently introduces artifacts, resulting in inconsistent green algae time series patterns. Hence, caution is warranted when jointly utilizing daily optical and SAR data, particularly for green tide simulation and forecasting purposes.

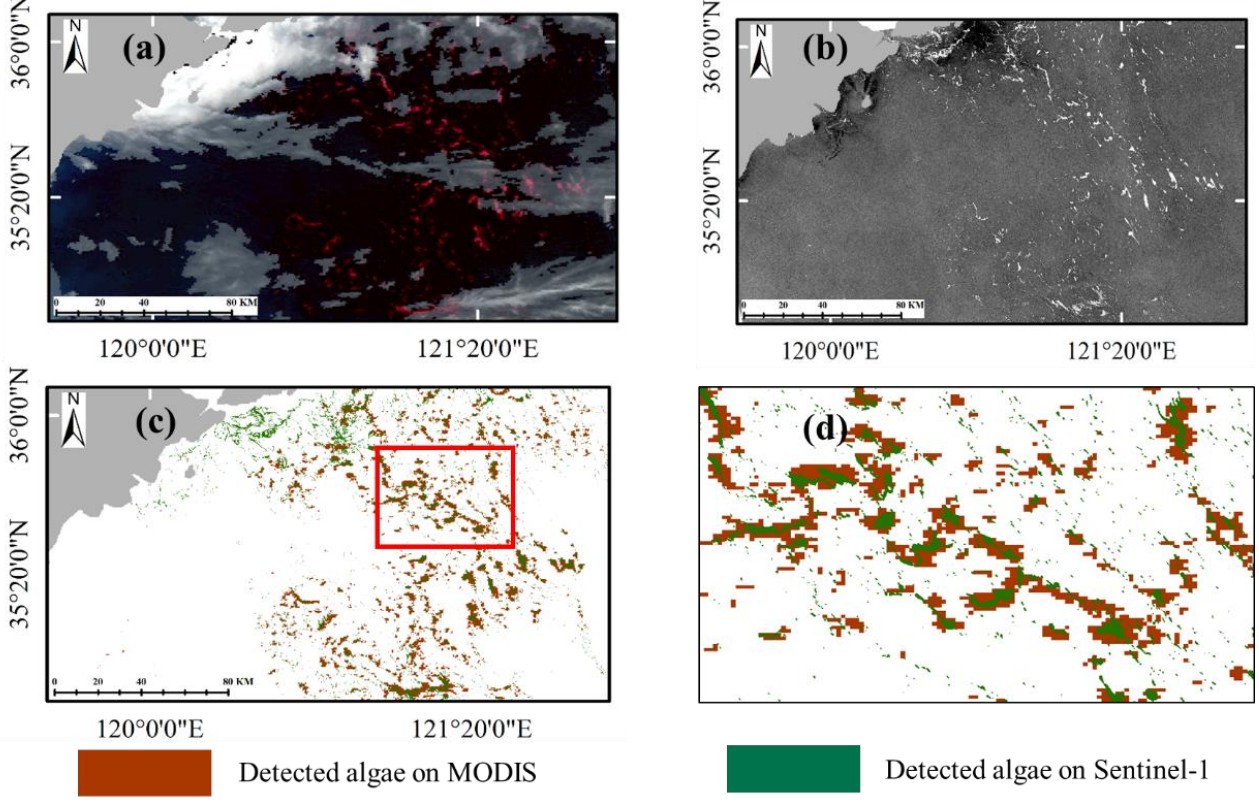

**Figure 6. Comparison of randomly selected optical and SAR image detection results (date: June 30, 2021).** (a) and (b) are the optical and SAR images observed on the same day; (c) are the overlapping green algae coverages devried from (a) and (b); and (d) is the zoom of partial area in (c).

To address this limitation and to generate a continuous and comprehensive green tide coverage dataset for the entire Yellow Sea, we present a fusion dataset consisting of weekly average green tide coverage data by merging the two datasets. Following the methodology outlined by Hu et al. (2023) for defining monthly green tide data, the specific process of integrating daily optical and SAR green tide products in this article is as follows:

1) Resample the daily SAR green tide coverage data, originally acquired at a resolution of 30 meters, to a resolution of 500 meters. This step ensures consistency with the resolution of the MODIS green tide product.

2) Our analysis indicates at least one valid daily green tide observation in the Yellow Sea every week. To integrate the daily optical and SAR green tide coverage products, we divide them into weekly intervals, forming a time unit consisting of daily data each week. We count the number of images (N) where green tides are observed within a given time unit without distinguishing between sensors. This count represents the total valid observations within the time unit.

3) For a specific pixel in the study area, we count the number of green algae occurrences (recorded as M), where M is less than or equal to N. When a pixel is identified as containing green algae by the AlgaeNet or GANet model, we assume it


contains 100% algae. Therefore, the proportion of algae in a pixel of the weekly product is calculated as M/N * 100%, and the corresponding coverage area of the pixel is M/N * 100% * 0.5 * 0.5 (in km2).

Taken June 17-23, 2019, as an example, throughout this week, there were seven daily optical and SAR green tide coverages, as depicted in the left image (Fig. 7a), while the right image displays the combined weekly green tide coverage (Fig. 7b),

following the data fusion steps mentioned above.

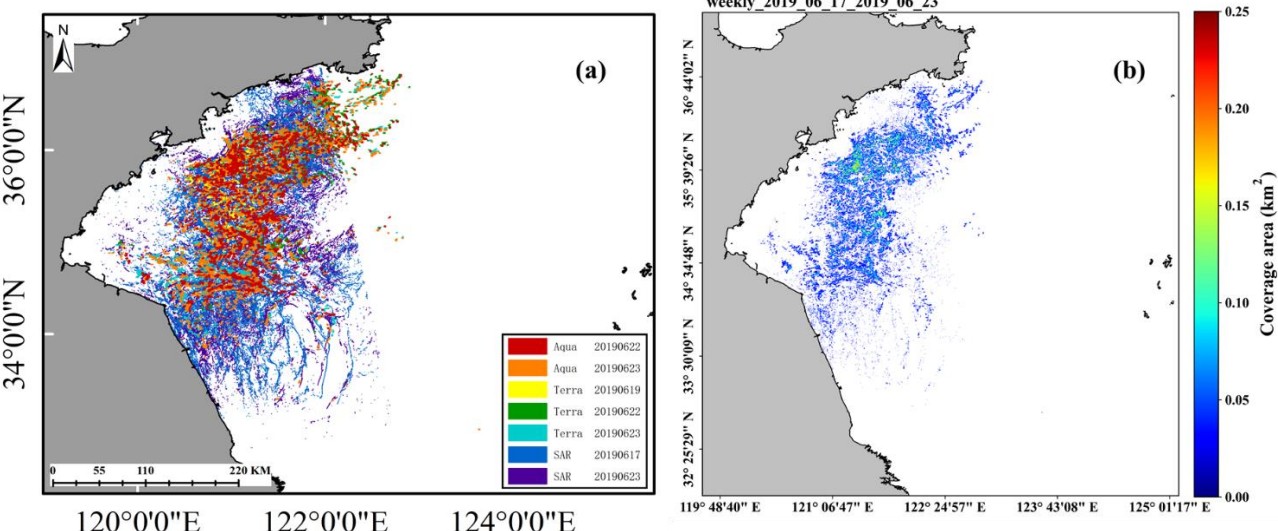

**Figure 7. Weekly integration sample of daily optical and SAR products from July 17-23, 2019.**

## 3 Results Validation and Discussion

Comparisons with other datasets are challenging due to the scarcity of internationally available similar datasets. The weekly product comes from the daily green tide product fusion. The data accuracy of the initial daily product directly affects the accuracy of the final fusion product. Therefore, the verification of the weekly product includes two parts. The first is verifying the daily product, and the second is verifying the final weekly fusion product.

### 3.1 Verification of daily green tide datasets in the Yellow Sea

Besides validating AlageNet and GANet models using the testing set, as described in the methodological section, it's crucial to comprehensively evaluate the daily green algae dataset extracted from the entire remote sensing image of the Yellow Sea region.

### 3.1.1 Daily MODIS product validation

For this study, MODIS images under cloud-free and thin cloud conditions on June 19, 2021, were randomly selected (Fig. 8).
The overall evaluation (mIOU) of the green tide data reached 85.86%, 69.81%, 84.23%, and 89.61%, respectively, for the entire Yellow Sea, open water, cloud area, and Subei Shoal area. Furthermore, through visual inspection of three randomly selected typical regions (Box 1-3), the green tide dataset demonstrated excellent accuracy (see Table 4 and Fig. 8).

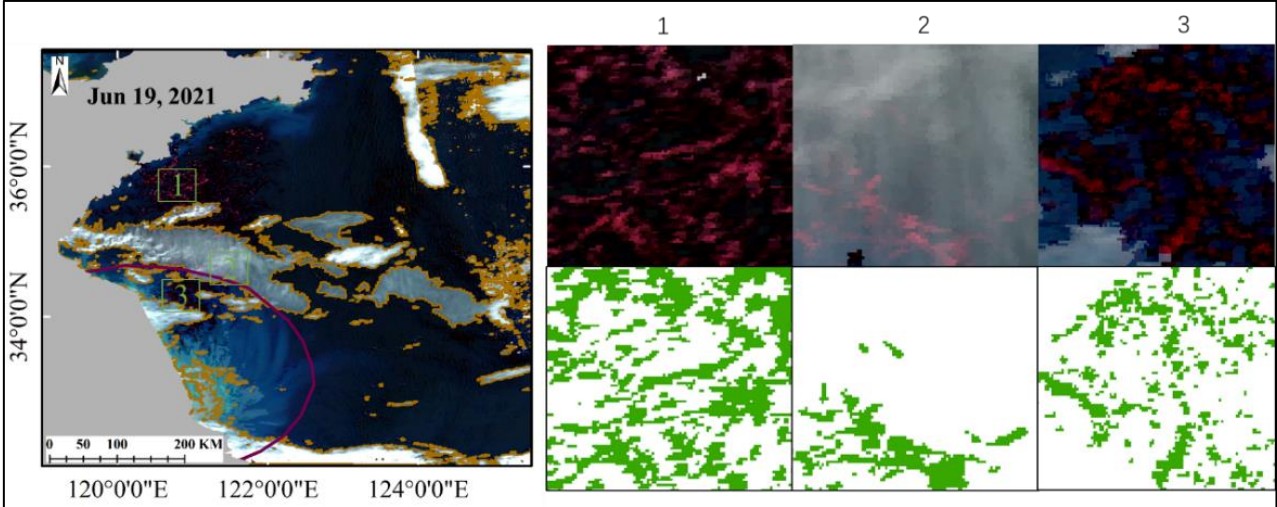

**Figure 8. Green tide detection result analysis from randomly selected optical images.** Box 1 is an enlarged view of the open sea area under cloud-free conditions; Box 2 is an enlarged view of the thin cloud area; Box 3 is an enlarged view of a portion of the nearshore area of Subei Shoal. (Date: June 19, 2021).

We categorize the entire algae tide outbreak process into two stages: the growth period (from satellite's initial algae coverage
to the largest coverage) and the dissipation period (from the largest algae coverage to complete disappearance). Considering the differences in imaging capabilities between the two optical satellites, Aqua and Terra, Figure 9 presents histogram statistics of the size of green algae strips detected from MODIS images. It was observed that during the growth stage, the size of green algae strips was primarily concentrated at <50 km$^2$, with the maximum algae strip reaching 400 km$^2$. In the dissipation stage, the size of algae strips was concentrated at <20 km$^2$, and the largest strip measured less than 150 km$^2$.
Theoretically, the lower detection limit for MODIS 250 m resolution bands is approximately 1% of the pixel size, i.e., 625 m$^2$ (Li et al., 2022b). However, based on the statistical results, the smallest detected strip of green algae (~1 km$^2$) is much larger than the theoretical threshold (625 m$^2$) due to resampling to 500 m resolution.




Open Access · Earth System · Science · Data · Discussions



**Figure 9. Statistical analysis of green algae patch size derived from optical imagery.**

### 3.1.2 Daily SAR product validation

In SAR images, the GANet model demonstrates its poorest detection capability in the shallow water area of the Subei Shoal (Guo et al., 2022). This deficiency is primarily attributed to the turbid seawater in the Subei Shoal, which reflects strongly on the SAR image, thereby reducing the contrast between seawater and green algae. Consequently, the accuracy of green algae extraction is lower compared to that in open sea areas. However, the new GANet model has significantly improved green algae detection capabilities by incorporating labeled samples from this area. As depicted in Fig. 10, the evaluation indices mIOU of a randomly selected green tide dataset in the Subei Shoal on June 22, 2018, reached 99.22%, showcasing reasonable performance (Fig. 10a-d). Moreover, for the MODIS image of the corresponding area on the same day (Fig. 10d-



e), green tide extraction also exhibits high accuracy, reaching 84.92%. The distribution pattern remains consistent for SAR

datasets (as shown in subfigures 10b and 10e). The relevant assessment of the entire image for the green tide dataset is

provided in Table 4. Additionally, SAR observation images covering the Yellow Sea were randomly selected (Fig. 10f-i),

where the overall mIOU reached 87.62%, further highlighting the excellent detection capability of the model.





Earth System
Open Access Science Discussions
Data


**Figure 10. Randomly selected SAR image detection results on the Subei Shoal (a-e) and the entire Yellow Sea (f-i) regions. (Date: June 22, 2018, and June 30, 2021).**

**Table 4. Green tide detection result analysis (random entire image examples)**

| Type | Area | Accuracy (%) | Precision(%) | Recall(%) | F1-score(%) | mIOU(%) |
|---|---|---|---|---|---|---|
| MODIS image on June 19, 2021 | Entire imagery | 99.72 | 93.23 | 91.57 | 92.39 | 85.86 |
| | Thin cloud area | 99.90 | 76.35 | 89.07 | 82.22 | 69.81 |
| | Subei Shoal | 99.95 | 93.15 | 89.79 | 91.44 | 84.23 |




| | | | | | | |
|---|---|---|---|---|---|---|
| | Open water | 99.79 | 96.71 | 92.42 | 94.52 | 89.61 |
| | Corresponding to SAR Subei Shoal area below | 99.82 | 96.36 | 87.73 | 91.84 | 84.92 |
| SAR image | Subei Shoal (on June 22, 2018) | 99.99 | 99.61 | 99.60 | 99.61 | 99.22 |
| | Yellow Sea (on June 30, 2021) | 99.84 | 99.85 | 87.73 | 93.40 | 87.62 |

**3.2 Verification of weekly green tide datasets in the Yellow Sea**

**3.2.1 Overall uncertainty verification**

The specific process is as follows:

  1) Utilize the fused weekly green tide data for a specific week to count the green algae coverage area for that week;

  2) Calculate the daily green algae coverage areas for each day within that week as S1, S2, ..., SK. If the total number of green tide observations in that week is K, the average green tide coverage area for that week is given by: (S1 + S2 + ... + SK)/K;

  3) Compare the weekly average green tide coverage area obtained from the fused dataset with the one calculated from the daily observations. Calculate the correlation and root mean square error (RMSE) between the two kinds of coverage area from 2008-2022.

Figure 11 indicates that compared to the second average green tide area used as the benchmark, the weekly green tide dataset in this article demonstrates a completely consistent result ($R^2$=1 and RMSE=0).

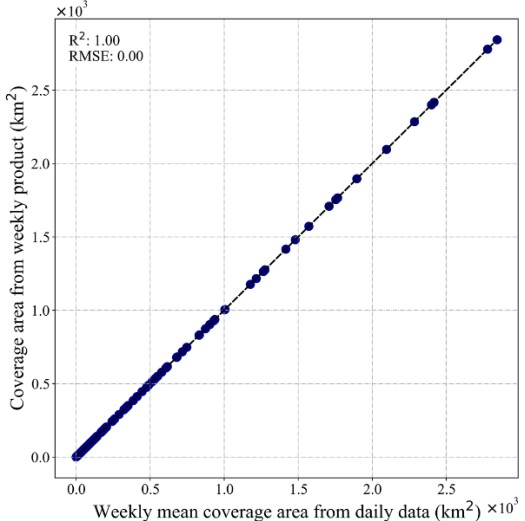

**Figure 11. Overall verification of weekly green tide datasets.**



### 3.2.2 Uncertainty verification of individual cases

To verify whether the weekly product conforms to the parabolic green tide outbreak pattern shown as Fig. 1d in each year, the specific process is as follows:

1). Generate Green Tide Outbreak Curve: Fit the outbreak parabolic curve of the entire life cycle of the green tide bloom using the satellite's daily green algae coverage data and the Gompertz curve model. The Gompertz curve model formula is shown below:

$$y = ae^{-be^{-cx}} \tag{1}$$

$y$ is the accumulative coverage of the satellite's daily green tide, $x$ is the corresponding date (Here it is expressed in terms of Day Of Year, DOY).$a$, $b$, and $c$ represent the fitted constant terms obtained by utilizing daily green tide coverage to fit the Gompertz curve model. We derived the life parabola curve by conducting piecewise fitting based on the green tide growth and dissipation periods.

2). Comparison with Daily Observations: Compare the daily green tide coverage obtained from satellite daily observations with the daily green tide coverage predicted by the parabolic curve.

3). Assessment of Daily Observation Uncertainty: Calculate the observation uncertainty ($R^2$ and RMSE, recorded as U1) of the satellite daily product by comparing it with the fitted parabolic curve.

4). Calculate Theoretical Weekly Average Green Tide Coverage: Based on the green tide parabolic curve, compute all
theoretically weekly average green tide coverage values throughout the entire outbreak cycle.

5). Assessment of Weekly Product Uncertainty: Assess the uncertainty ($R^2$ and RMSE, recorded as U2) between the weekly product in this article and the theoretically weekly average data obtained from the parabolic curve.

6). Comparison of Weekly and Daily Uncertainty: Evaluate the difference level between the uncertainty of the weekly product and the daily uncertainty, i.e., U1 v.s. U2.

We randomly chose the green tide outbreak in 2019 as a case study, as demonstrated in Fig. 12. The fused weekly green tide data not only conforms to a parabolic outbreak trend but also its variance is close to the theoretical outbreak curve with an $R^2$ value of 0.89 and an RMSE of 275 km$^2$ (U2, see Fig. 12b). Moreover, Figure 12a shows the uncertainty associated with the weekly data exhibits a stronger correlation and smaller RMSE value than the daily data, with an $R^2$ value of 0.61 and an RMSE of 794 km$^2$ (U1). Furthermore, we noted a decrease in the uncertainty of the fused weekly product, corresponding to
an increase in the effectiveness of daily satellite observations over the week.




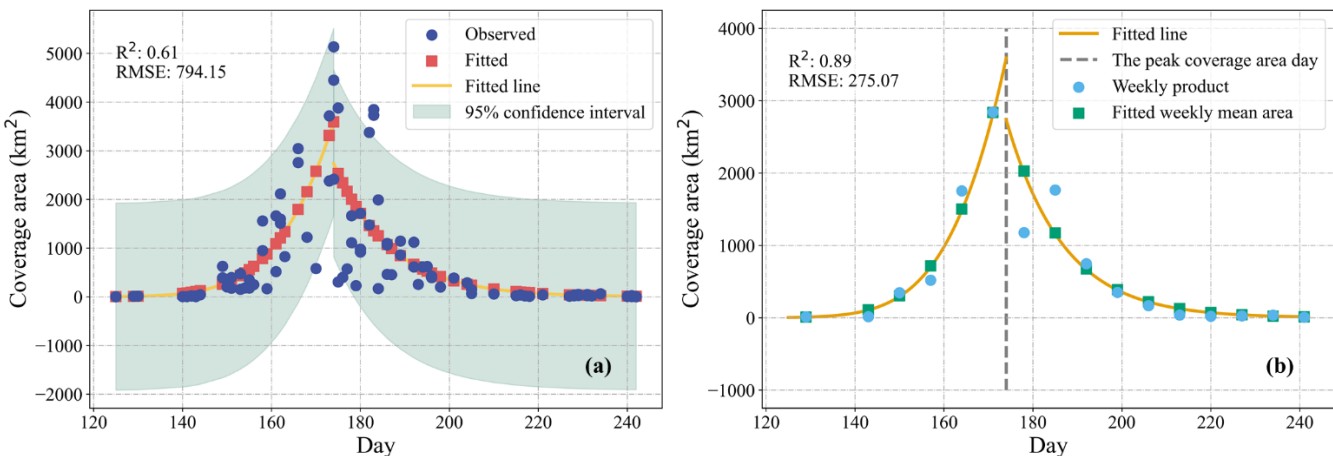

**Figure 12. Uncertainty assessment of green tide products based on green tide outbreak pattern (2019).** (a) Uncertainty in daily product and (b) Uncertainty in weekly product.

## 3.3 Spatial characteristics of green tide coverage.

In Fig. 13, the frequency of algae-containing pixels over the years is depicted, accompanied by a hotspot map illustrating the distribution of green algae. The analysis reveals that over 90% of green algae occurrences are concentrated in the shallow waters of the central Yellow Sea (water depth ≤ 30 m). This region exhibits higher water transparency than the shallow coastal waters off Subei Shoal, facilitating enhanced light absorption by floating algae during their drift, thereby promoting their growth. Additionally, the area surrounding the Shandong Peninsula experiences a notable influx of green algae (highlighted in the black box in Fig. 13), attributed primarily to the influence of wind forces and ocean currents.

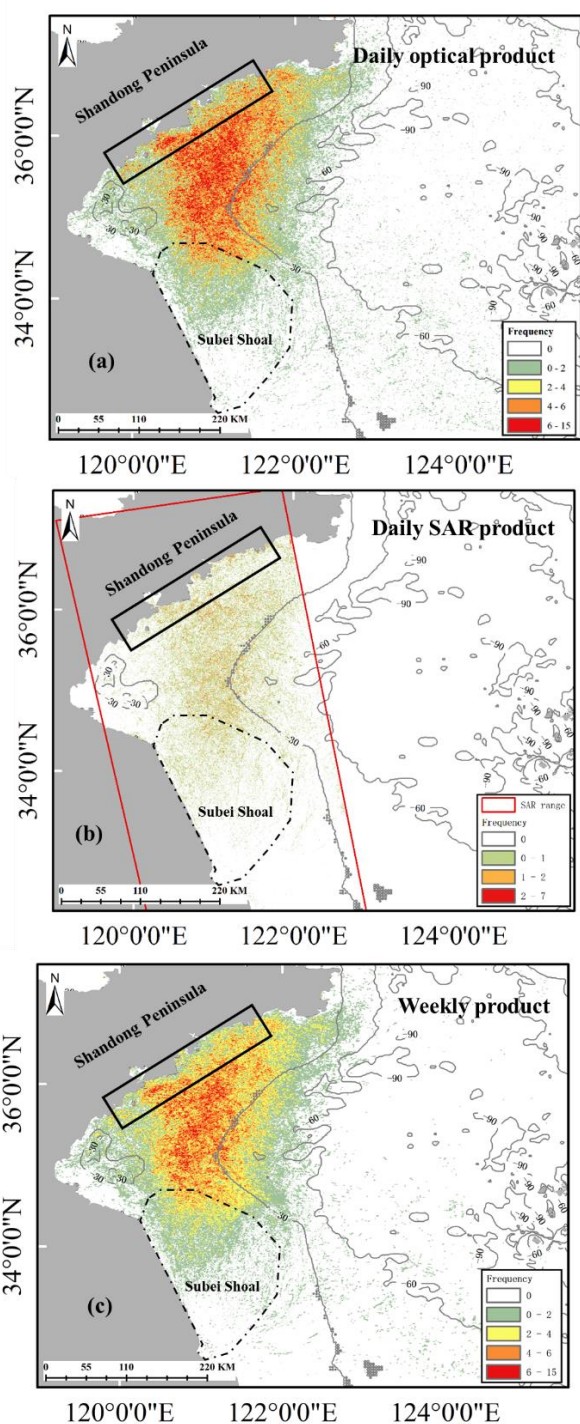

**Figure 13. Green tide distribution hotspot map obtained from daily optical (a) and SAR (b) products, and weekly products (c).**



### 3.4 Temporal characteristics of green tide coverage

Based on daily and weekly green tide datasets derived from optical and SAR images, we analyzed to determine the annual maximum area coverage of algae observed by both satellites. The resulting maximum coverage time series aligns with findings from Cao et al. (2023) and Hu et al. (2023). As shown in Fig. 14a, from 2008 to 2012, there was a notable downward trend in annual variations of green tide coverage. However, since 2012, there has been a rapid expansion in the scale of green tide outbreaks, particularly evident since 2018. Notably, the outbreak in 2019 reached unprecedented levels,

partially influenced by proposed prevention and control measures. However, these measures have not been entirely effective in curbing large-scale green tide outbreaks (Feng et al., 2020; Hao et al., 2020; Sun et al., 2022). It's worth noting that SAR sensors only detect green algae completely exposed on the sea surface (as depicted in Fig. 1c), resulting in slightly smaller coverage areas compared to daily optical detection (indicated by the red polyline in Fig. 14a). This discrepancy can also be attributed to variations in the dates corresponding to the maximum coverage areas monitored by the two satellites. For

instance, in 2021, the maximum daily coverage on MODIS images occurred on June 23, while the maximum daily green tide coverage on SAR images was recorded on June 12. Furthermore, analysis in Fig. 14b highlights 2012 and 2019 as the years with the lowest and largest values in the historical green tide coverage time series. Notably, the onset of a large-scale green tide outbreak typically occurs much earlier than periods without such outbreaks. Therefore, the scale of a green tide outbreak is directly proportional to its initial onset time. Earlier observations of green algae by satellites correspond to larger expected

outbreak scales. Additionally, the final scale of the green tide outbreak is also directly proportional to the initial outbreak scale (Cao et al., 2023).





**Figure 14. (a)Time series of the maximum coverage area of the green tide from 2008 to 2022 and (b) green algae coverage in 2012 and 2019.**



## 3.5 The mechanism and impact of green tide outbreaks

Variations in green tide coverage encompass both physical drift and diffusion processes alongside biological proliferation, which is influenced by environmental factors and reciprocally affects environmental conditions. Originating from the Subei Shoal (Fig. 1a-b), the green tide drifts from south to north into the central Yellow Sea driven by currents and wind fields, corroborated by dynamic time series data from optical MODIS and Sentinel-1 SAR (see supplementary animation). Previous research indicates the scale of green tide outbreaks is linked to *Porphyra* mariculture along the Subei Shoal coast (Xing et al., 2019; Cao et al., 2023). Green algae spores released from mariculture rafts can trigger large-scale green tide events under favorable conditions. Analysis of Sentinel-1 SAR imagery (Fig. 15) reveals a declining trend in mariculture areas from 2019 to 2021, yet significant green tide outbreaks still occurred in those years. Consequently, predicting green tide outbreaks solely based on mariculture raft areas is unreliable, as the precise number of spores entering the seawater cannot be accurately determined.

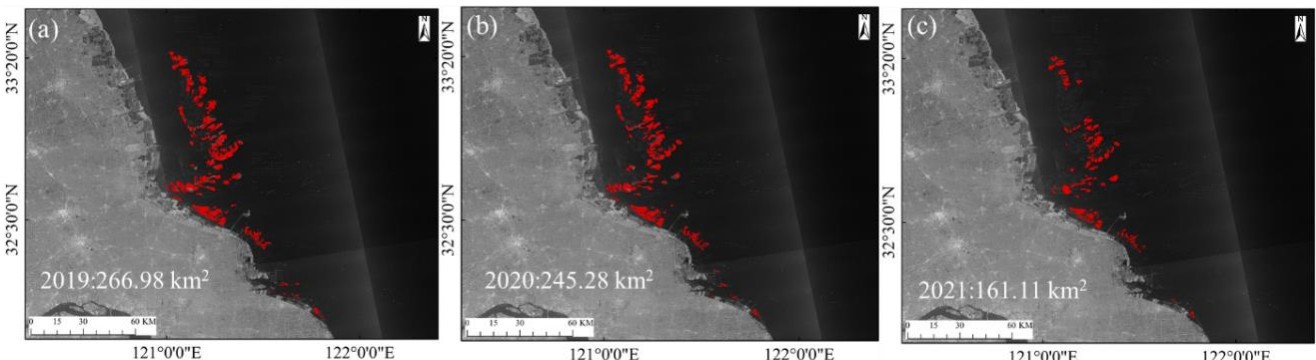

**Figure 15. Analysis of changes in the *Porphyra* mariculture region in the Subei Shoal area. Its position corresponds to Figure 1a-b.**

In addition to the influence of *Porphyra* mariculture, environmental factors such as SST, SSS, SSC, SSW, O2, and nutrients, including the derived gradients of these key elements, play crucial roles in the growth and dissipation of green tide. Recognizing the interdependency among these environmental elements, we leveraged the advantages of a deep-learning-based model to conduct a multi-factor collaborative analysis, establishing a mapping model between these factors and daily green tide coverage. As illustrated in Fig. 16, we utilized the XGBoost model to correlate environmental elements with daily green tide coverage. The Permutation Importance and SHAP methods based on the XGBoost model were employed to rank the importance of each environmental element influencing green tide changes. Our findings indicate that O2 holds the highest rank in importance, followed by SSS and nutrients, particularly nitrate. Additionally, the derived gradients of SSS, O2, and SST also ranked high in importance. O2 in the Yellow Sea exhibits a declining trend throughout the green tide's life cycle from May to August yearly (Fig. 17a). During the growth period, the initial high concentration of O2 negatively correlates with the rapid growth of green tide coverage (Fig. 16c).





In contrast, during the dissipation period, the decreasing O2 concentration positively correlates with the rapid dissipation of green algae (Fig. 16f). Similarly, nutrient levels, especially nitrates, in the Yellow Sea show an initial increase followed by a downward trend from May to August each year (Fig. 17b). Nutrient concentration consistently correlates positively with

changes in green tide coverage throughout its entire lifecycle (Figures 16c and 16f). Thus, the proliferation process of green tides significantly influences the water quality of the Yellow Sea, consuming substantial amounts of O2 and nutrients and leading to clearer water conditions after September. The concentration of O2 and nutrients in different years partly determines the scale of green tide outbreaks (Wang et al., 2023b). For instance, the smaller scale of the green tide outbreak in 2020 compared to 2019 and 2021 can be attributed to lower nitrate concentrations (see Fig. 1d and Guo et al., 2022).

Therefore, the combined action of green tide proliferation and drift processes results in significant yearly variations in the scale of green tide outbreaks.

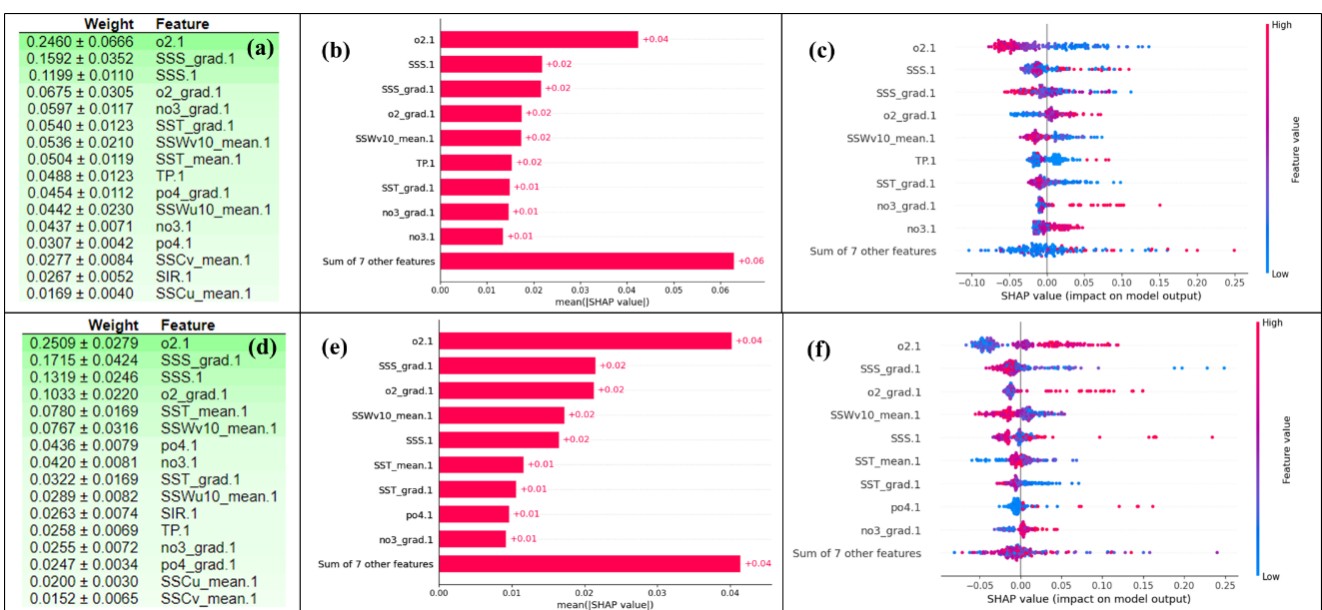

**Figure 16. Importance ranking of main environmental factors in the growth (a-c) and dissipation (d-f) stages.** Subfigures (a) and (d)

the Permutation Importance; Subfigures (b) and (e) the Importance of SHAP; and Subfigures (c) and (f) the correlation between environmental factors and green tide coverage based on SHAP theory.



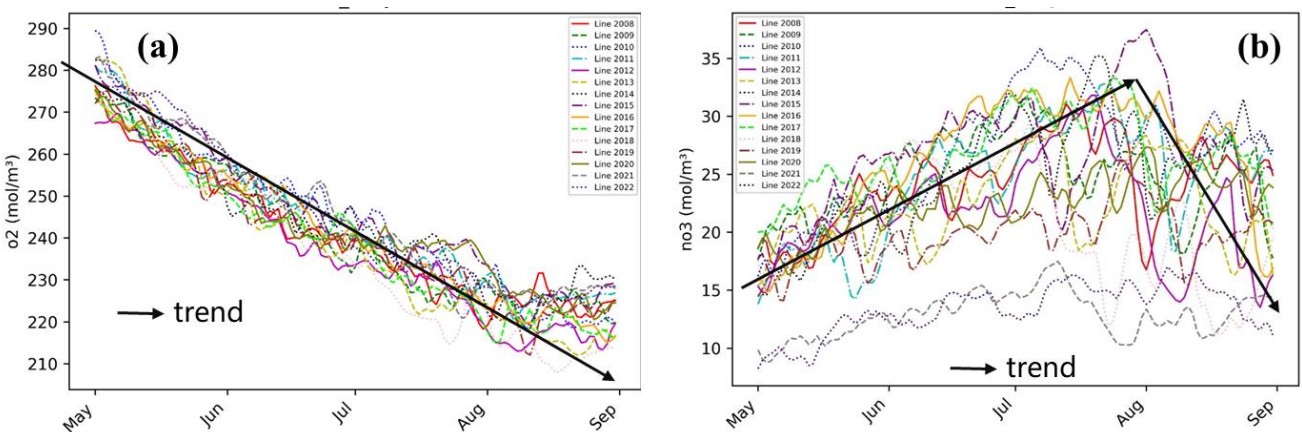

**Figure 17. Changes in marine dissolved oxygen (a) and nitrate concentrations (b) during the green tide outbreak from 2008 to 2022.**

## 470  4 Code and data availability

**Weekly green tide products and deep-learning models in this study.** The following datasets and models will be released, as detailed in Table 3:

1) The study releases the weekly green tide coverage datasets by integrating optical MODIS and Sentinel-1 SAR images spanning the periods of 2008-2022 with 500 m spatial resolution, which can be used as foundational data for green

tide model simulation and forecasting; the weekly product conforms to the life pattern of green tide outbreaks and exhibits parabolic curve-like characteristics, with an uncertainty of $R^2$=0.89 and RMSE=275 $km^2$. The weekly green tide product is provided in Tagged Image File (TIF) format.

2) The continuous weekly datasets were derived from daily green tide datasets, and what the AI-based AlgaeNet and GANet model directly detects is also daily green tide coverage, therefore daily green tide dataset with 500m spatial

resolution from MODIS images and 30m spatial resolution from SAR images are also released as verification datasets for other green tide extraction models (e.g., Zhou et al., 2021; Qi et al., 2022a; Wang et al., 2023); the overall indices demonstrated a higher mIOU, reaching 85.86% and 87.62%, respectively, based the optical and SAR images covering the entire Yellow Sea. The daily green tide product is provided in shapefile format.

3) The annotated green tide sample dataset will also be shared as ground truth. This ground truth dataset can be utilized

as a training and testing set for developing other AI-based green algae extraction models. Optical green algae labels from MODIS images with 500 m spatial resolution and the green algae labels from SAR images with 30 m spatial resolution are provided in Portable Network Graphic (PNG) format.

4) The retrained AI-based green tide detection models—AlgaeNet and GANet. The models achieved a comprehensive metric mIOU of 67.51% (85.41%) based on optical (SAR) testing labels. These models offer scalable and effective tools for

future green tide extraction tasks. Furthermore, they can serve as pre-trained models for analyzing various satellite imagery types, including optical GOCI and microwave Gaofen-3 imagery.



**Table 3. Shared green tide related data products**

| Product | Type | Dataset description | Format | Temporal resolution | Spatial resolution |
|---|---|---|---|---|---|
| Green Tide Coverage Products | Optical | Daily green tide coverage under cloudless conditions | .shp | daily | 500 m |
| | SAR | Daily green tide coverage | | | 30 m |
| | Fusion products | Continuous weekly average green tide coverage | .tif | weekly | 500 m |
| Green Tide Annotation Sample Dataset | Optical | Green wave annotation sample set of optical images | .png | daily | 500 m |
| | SAR | Green tide annotation sample set of SAR images | | | 30 m |
| Green Tide Detection Program | Optical | AlageNet model for optical imagery | .py | / | |
| | SAR | GANet model for SAR imagery | | | |

The dataset is openly accessible to the public without any restrictions. It is permanently stored at http://dx.doi.org/10.12157/IOCAS.20240410.002 (Gao et al., 2024), where green tide coverage dataset is available as separate daily and weekly files. Both the daily and weekly datasets utilize the WGS84 spatial reference system and the UTM51 projected system.

Additionally, the green tide annotation sample dataset and the code for the green tide detection program can be freely
accessed at the same website, providing convenient access to annotated samples and code for algae detection programs. All code is written in Python.

**5 Conclusions**

The Yellow Sea, located in the northwest Pacific, has been witnessing an ecological anomaly known as the green tide since 2008. This phenomenon, characterized by the rapid proliferation and accumulation of large floating algae, has escalated into
one of the world's largest-scale marine disasters caused by green algae blooms, attracting significant international attention. Satellite remote sensing has emerged as the primary data source for detecting occurrences of the green tide, benefiting from its advantages in acquiring data with full coverage, high frequency, and periodic monitoring capabilities. The rapid advancement of artificial intelligence (AI) technology in recent years has provided significant advantages over traditional



methods for the precise and intelligent extraction of green algae in satellite imagery. In this study, we utilize optimized and scalable AI models, namely AlgaeNet and GANet, to conduct a comprehensive extraction and analysis of the Yellow Sea green tide. Our analysis encompasses optical MODIS images (500 m resolution) from 2008 to 2022 and microwave Sentinel-1 Synthetic Aperture Radar (SAR) images (30 m resolution) from 2015 to 2022, with a temporal resolution of one day. The evaluation of green algae extraction achieved notable results with the two models, reaching a comprehensive evaluation index (mIOU) of 67.51% and 85.41% based on 662 and 267 pairs of optical and SAR-labelled testing samples, respectively. Moreover, when applied to randomly selected optical and SAR entire images of the Yellow Sea, the indices demonstrated even higher mIOU, reaching 85.86% and 87.62%, respectively. Most importantly, we present a continuous and seamless dataset of weekly average green tide coverage. This weekly dataset is derived from integrating daily optical and SAR green tide coverage during each week of the green tide breakout. Through verification, the overall assessment of the uncertainty of this weekly product shows it is completely consistent with the overall direct average of the daily product ($R^2=1$ and RMSE=0). Additionally, the individual case verification of the green tide outbreak in 2019 also shows that the weekly product conforms to the life pattern of green tide outbreaks and exhibits parabolic curve-like characteristics, with an uncertainty of $R^2=0.89$ and RMSE=275 km$^2$. This weekly green tide dataset provides an independent and reliable source of long-term data spanning 15 years in the Yellow Sea, facilitating comprehensive research in various domains such as forecasting, numerical model simulation, regional and national-scale climate change analysis, and formulation of disaster prevention plans.

**Supplement.** An animation of weekly green tide datasets from 2008 to 2022 has been uploaded as supplementary material.

**Author contributions.** Le Gao and Xiaofeng Li designed the study, and Yuan Guo and Le Gao developed the deep learning code for the AlgaeNet and GANet models and final green tide datasets. Yuan Guo processed the optical and SAR data and performed the computations. All authors discussed and contributed to the models, datasets, and manuscript.

**Competing interests.** The authors declare no competing interests.

**Acknowledgments.** This study was jointly supported by the National Natural Science Foundation of China (42376175, U2006211 and 42090044) and the Strategic Priority Research Program of the Chinese Academy of Sciences (XDB42040401).

The MODIS surface reflectance products can be accessed through the following URLs: MYD09GA (https://ladsweb.modaps.eosdis.nasa.gov/missions-and-measurements/products/MYD09GA) and MOD09GA (https://ladsweb.modaps.eosdis.nasa.gov/missions-and-measurements/products/MOD09GA).

The European Space Agency (ESA) and NASA provided the Sentinel-1 SAR images via https://scihub.copernicus.eu and https://search.asf.alaska.edu/.



Daily sea surface temperature (SST), sea surface salinity (SSS), and sea surface circulation (SSC) data were sourced from

HYCOM data (https://developers.google.com/earth-engine/datasets/catalog/HYCOM_sea_water_velocity and

https://developers.google.com/earth-engine/datasets/catalog/HYCOM_sea_temp_salinity). Daily precipitation and sea

surface wind (SSW) at 10m data were provided by the fifth-generation atmospheric reanalysis data (ERA5) of ECMWF

(https://developers.google.com/earth-engine/datasets/catalog/ECMWF_ERA5_DAILY).

Nutrients (such as phosphates (PO4) and nitrates (NO3)) and dissolved oxygen (O2) data with a resolution of 0.25° were

obtained from CMEMS/Global Ocean Biogeochemistry Analysis and Forecast

(https://data.marine.copernicus.eu/product/GLOBAL_MULTIYEAR_BGC_001_029/description and

https://data.marine.copernicus.eu/product/GLOBAL_ANALYSIS_FORECAST_BIO_001_028/description). Solar radiation

(SIR) with a resolution of 0.25° was retrieved from the GSFC data portal

(https://neo.gsfc.nasa.gov/view.php?datasetId=CERES_INSOL_M).

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
