# Peer review of "Weekly Green Tide Mapping in the Yellow Sea with Deep Learning: Integrating Optical and SAR Ocean Imagery"

_Earth System Science Data, 2024_

## Author Comment (AC1)

**Response to Reviewer's Comments**

There is a long history of floating green macroalgae blooms in the Yellow Sea, which might be dated back to the year of 1999 (Lline 130, the introduction is wrong). On the basis of optical and microwave data of MODIS, Sentinel-1, the authors presented a weekly dataset of green tide coverage in the Yellow Sea since 2008. With the consideration of the continuing occurrence of large scale green tides in the Yellow Sea in recent years, the dataset in this paper would be helpful for exploring the dynamics of the green tide as well as the causes. This paper may be published after making some improments listed as below.

At Line 130, we revised the recorded timing of the macroalgae observations to: 'Satellite records of the Yellow Sea green tide date back to 1999, with large-scale outbreaks being observed since 2008.' The manuscript was subsequently modified and revised according to the reviewers' comments.

1. For the dataset, why the weekly one is important when the daily and monthly dataset are available (Hu et al., 2023)? No biomass is presented or discussed in the paper, which is a drawback. Please clarify these issues. And, the time series of maximum daily coverages and weekly coverages should be presented in the results.

    (a) Why the weekly product is important?

    Due to cloud and rain interference, daily optical MODIS satellite records of green tide coverage contain numerous missing values. The daily Sentinel-1 SAR satellite has a relatively slow return cycle of 6 days, resulting in low observation frequency for daily green tide monitoring. The green tide outbreak process is characterized by rapid changes over short periods. Therefore, daily optical and SAR data cannot timely and accurately capture the outbreak process at a continuously high frequency.

    To address the limitations of daily data, Hu et al. (2023) proposed using monthly average green tide coverage data. However, the Yellow Sea green tide occurs from early May to the end of August, lasting about four months. Consequently, the monthly average data provides only one records per month, totaling four records for the entire life cycle, which is insufficient to fully describe the outbreak dynamics due to its low temporal resolution.

    In response to the shortcomings of daily and monthly records, we proposed a weekly average green tide coverage dataset. This dataset offers higher temporal resolution than monthly averages and provides seamless and continuous coverage compared to daily data, making it more suitable for studying green tide outbreaks.

    To this end, we further emphasized the importance of weekly products in the Abstract and Introduction sections.

    (b) Biomass

    The biomass of the green tide is calculated by multiplying the detected algae pixel coverage area by a calibration constant derived from water tank and in situ experiments (as shown in formula 3 "$T=S\times\sigma_0$"of Hu et al., 2019). Since biomass is linearly related to the green tide coverage area, identifying algae pixels and measuring their coverage area are fundamental tasks. Consequently, the dataset released in this paper is a green tide coverage dataset.

    (c) Time series of maximum daily coverages and weekly coverages

The time series of maximum daily and weekly coverages is shown in Fig. 15 in Section 3.4. Weekly coverage data is calculated from the daily coverages within a week, with the default being the middle of each week. The results indicate that the weekly maximum positioned between the daily maximum coverages observed by optical and SAR satellites, as shown in Fig. 15a. Additionally, due to different revisit cycles and varying effective observation frequencies under cloud and rain cover, the maximum coverage area of daily green tide observed by optical and SAR satellites occurs at different times, as shown in the 2019 time series in Fig. 15b. By analyzing the time of the first green tide observation and the maximum coverage area observed, we can predict and evaluate the scale of the green tide outbreak for that year.

[Figure]

Figure 15. (a) Time series of the maximum daily and weekly coverage area of the green tide from 2008 to 2022, and the numbers on the subfigure represent the corresponding observation date and the week to which the fusion product belongs; (b) daily and weekly green algae coverage throughout the entire life cycle in 2012 and 2019.

Hu, C., Qi, L., Hu, L., Cui, T., Xing, Q., He, M., Wang, N., Xiao, Y., Sun, D., Lu, Y., Yuan, C., Wu, M., Wang, C., Chen, Y., Xu, H., Sun, L.E., Guo, M., Wang, M., 2023. Mapping Ulva prolifera green tides from space: a revisit on algorithm design and data products. Int. J. Appl. Earth Obs.

Geoinf. 116, 103173 https://doi.org/10.1016/j.jag.2022.103173.

Hu, Lianbo, Kan Zeng, Chuanmin Hu, and Ming-Xia He. 2019. "On the Remote Estimation of *Ulva Prolifera* Areal Coverage and Biomass." Remote Sensing of Environment 223: 194–207. https://doi.org/10.1016/j.rse.2019.01.014.

2. For the growth model of green tide, the discussion seems to be misleading. As shown by Xing et al., 2018, 2019, the maximum biomass (or coverage area) of green tide is not only regulated by the initial biomass on the seaweed cultivation rafts but also mainly the lasting period in the eutrophicated turbid waters of the Jiangsu shoal where the green macroalgae has a high daily growth rate. The lasting period is regulated by the patterns of wind and sea surface current, for which I am not sure that the HYCOM would provide the accurate data over the study area. The biomass of green tide in 2023 give a good case on the growth model of green tide. The authors may check and revise the relevant sections of this paper. I suggest to remove the section 3.5.

Although HYCOM environmental factor data have been used in some studies on the mechanism of green tide outbreaks (e.g., Jin et al., 2018; Li et al., 2021a and 2021b), the quality of HYCOM data in the shallow waters of the Yellow Sea requires further optimization and verification. To avoid drawing potentially unreliable conclusions, we removed Section 3.5 and other related paragraphs.

Jin, S., Liu, Y., Sun, C., Wei, X., Li, H., and Han, Z.: A study of the environmental factors influencing the growth phases of Ulva prolifera in the southern Yellow Sea, China, Mar. Pollut. Bull., 135, 1016-1025, https://doi.org/10.1016/j.marpolbul.2018.08.035, 2018.
Li, D., Gao, Z., and Song, D.: Analysis of environmental factors affecting the large-scale long-term sequence of green tide outbreaks in the Yellow Sea, Estuarine Coastal Shelf Sci., 260, 2021a.
Li, D., Gao, Z., and Xu, F.: Research on the dissipation of green tide and its influencing factors in the Yellow Sea based on Google Earth Engine, Mar. Pollut. Bull., 172, 2021b.

3. The species of green tide is Ulva prolifera. Pelase correct the term.

The term "Enteromorpha" has been changed to "Ulva prolifera."

4. The overflying time of different sensors and the impacts on the results should be presented and discussed.

We conducted a spatiotemporal analysis and discussion on the overflying times of MODIS and SAR sensors and the corresponding green tide coverage results. This discussion is supplemented in the manuscript;

(a) Figure 6 shows the green algae pixels detected by the MODIS and SAR sensors on the same day. The distribution patterns of the green algae patches obtained by the two sensors are very consistent. However, due to the higher resolution of SAR images, many tiny algae patches are detected by the Sentinel-1 SAR sensor but not by the optical MODIS sensor. Additionally, since the MODIS sensor has some underwater detection capability, while SAR can only observe algae patches completely floating on the sea surface, the algae strips detected optically are wider. One of the purposes and necessities of this study is to use the two sensors to complement each other and generate a more accurate green tide coverage by fusing the daily algae coverage detected by both

optical and SAR sensors (see Section 2.4).

[Figure]

Figure 6. Randomly selected algae distribution patterns observed by MODIS and SAR sensors on the same day (June 23, 2019).

(b) Figure 7 shows the optical remote sensing images and green tide coverage results obtained by MODIS Aqua and Terra sensors under cloudless conditions on June 23, 2019. Due to the different overflying times of the Aqua and Terra sensors, one image was obtained in the morning (02:15 UTC for Terra) and the other in the afternoon (5:30 UTC for Aqua), with a time interval of about 3 hours. When the satellite sensors operate and observe the sea surface, the final quality of the optical images obtained by the Aqua and Terra sensors varies due to different environmental factors such as sunglint, atmospheric refraction, and solar inclination. Correspondingly, the size of the green tide coverage area detected by the proposed deep learning model is different (e.g., 5795.40 km² vs. 4789.74 km² on June 23, 2019). Similarly, SAR images exhibit this phenomenon compared to optical sensors. To mitigate the impact of overflying time on the green tide detection results on the same day, averaging the green tide coverage obtained by two or more sensors at different overflying times is a good approach. This is also one of the advantages of the weekly product over daily green tide coverage (see Section 2.4).

[Figure]

Figure 7. Randomly selected observation images and extraction results of Aqua and Terra on the same day (June 23, 2019).

(c) Due to interference and obstruction from clouds and rain and the different revisit periods, effective observation frequencies of the two different satellites, optical MODIS and Sentinel-1 SAR, vary, resulting in different area sizes and corresponding times for the maximum daily green tide coverage obtained. Figure 8 shows a randomly selected example of the maximum daily green tide coverage obtained by the two sensors, MODIS/(Aqua & Terra) and Sentinel-1 SAR, during the green tide bloom in 2021. Weekly averaging eliminates the differences in maximum green tide coverage caused by overflying time, which is another advantage of the weekly product over daily data (see Section 2.4).

[Figure]

Figure 8. The maximum daily green tide coverage obtained by the different sensors, MODIS/(Aqua & Terra) and Sentinel-1 SAR.

(d) Figure 11 shows the statistical characteristics of green algae strips of different sizes. The largest green algae strip has an area of >400 km², while the smallest has an area of only 1 km². Satellites are often more capable of observing large green algae strips but less capable of detecting smaller ones. However, small green algae strips are far more numerous. Therefore, by averaging multiple observations over different overflying times, the differences in observations of smaller strips can be mitigated to a certain extent. For example, Terra detected more small green algae patches due to favorable observation conditions, while Aqua missed many small strips due to poor observation quality. These differences over different overflying times can be partially overcome through weekly averaging.

[Figure]

Figure 11. Statistical analysis of green algae patch size derived from optical imagery.

5. The verification in the section of 3.2.1, is not useful, and may be removed.

The section 3.2.1 was removed.

6. Fig.13 subfigures should be presented with a same granule size.

Figure 13b (Figure 14b of the new manuscript) has been redrawn to match the granule sizes of figures a and c.

[Figure]

Figure 14. Green tide distribution hotspot map obtained from daily optical (a) and SAR (b) products, and weekly products (c).

7. The abstract does not mention the discontinuity in the daily coverage data. The "continuity" of the weekly data versus the daily data is a significant difference and should be emphasized.

We further revised the abstract to mention the discontinuity in the daily coverage data and highlight the "continuous" and "seamless" nature of the weekly product. The abstract is as follows:

"Since 2008, the Yellow Sea has experienced a world's largest-scale marine disasters, known as the green tide, marked by the rapid proliferation and accumulation of large floating algae. Leveraging advanced artificial intelligence (AI) models, namely AlgaeNet and GANet, this study comprehensively extracted and analyzed green tide occurrences using optical Moderate Resolution Imaging Spectroradiometer (MODIS) images and microwave Sentinel-1 Synthetic Aperture Radar (SAR) images. **However, due to the interference of clouds and rain, the daily green tide data often have large gaps, resulting in discontinuity, which limits their use.** Therefore, this study presents a continuous and seamless weekly average green tide coverage dataset with the resolution of 500 m, by integrating high precise daily optical and SAR data for each week during the green tide breakout. The uncertainty assessment shows that this weekly product conforms to the life pattern of green tide outbreaks and exhibits parabolic curve-like characteristics, with an low uncertainty ($R^2$=0.89 and RMSE=275 $km^2$). This weekly dataset offers reliable long-term data spanning 15 years, facilitating research in forecasting, climate change analysis, numerical simulation and disaster prevention planning in the Yellow Sea. The dataset is accessible through the Oceanographic Data Center, Chinese Academy of Sciences (CASODC), along with comprehensive reuse instructions provided at http://dx.doi.org/10.12157/IOCAS.20240410.002 (Gao et al., 2024)."

8. Regarding GANet, what is the rationale for selecting specific GLCM features? Since GLCM texture includes many features, which ones are most important for green tide detection?

Through ablation experiments, we found that Mean, ASM (angular second moment), and Entropy—three texture features expressed by the gray-level co-occurrence matrix (GLCM) obtained from SAR images—play a more critical role in the extraction of green tide. This is also partially supported by the findings of Hall-Beyer, 2017 and Liu et al., 2015. These three SAR texture images have been shared in the green tide dataset. The related statement has been added to the section 3.2.3.

Hall-Beyer., M.: Practical guidelines for choosing GLCM textures to use in landscape classification tasks over a range of moderate spatial scales, Int. J. Remote Sens., 38, 5, 1312–1338, 2017, doi: 10.1080/01431161.2016.1278314

Liu, H., Guo, H., and Zhang, L.: SVM-based sea ice classification using textural features and concentration from RADARSAT-2 dual-pol ScanSAR data, IEEE J. Sel. Topics Appl. Earth Observ. Remote Sens., 8, 4, 1601–1613, 2015, doi:10.1109/jstars.2014.2365215.

9. Lines 49-50: The phrase "Unaffected by clouds and rain, ... by clouds and rain" is repeated and should be revised.

The sentence was revised to "Unaffected by clouds and rain, SAR sensors operate under all-weather conditions".

10. In Figure 2 on page 6, MODIS is mentioned for optical images. It is recommended to specify Sentinel-1 SAR images when discussing SAR images.

The original Fig. 2 has been modified as follows:

[Figure]

Figure 2. Overall flow chart of green tide coverage products generation.

11. In line 226, the phrase "when feeding image slices into the AlageNet model" appears in a paragraph discussing the GANet model. Please clarify the intended model. Based on the context, it seems this should refer to the GANet model, not the AlageNet model.

Yes, this refers to the GANet model, not the AlgaeNet model. We have replaced AlgaeNet with GANet.

12. The authors developed two models, AlgaeNet and GANet. Why not create a unified model to identify green algae using both SAR and MODIS images?

The AlgaeNet model itself has the ability to detect green algae in both optical and SAR images (Gao et al., 2022). However, based on the U-Net architecture, we added attention mechanisms, sample imbalance loss functions, and texture feature enhancement information and further proposed the GANet model. Compared with the AlgaeNet model, the proposed GANet model has higher green algae extraction accuracy for SAR images (Guo et al., 2022). Therefore, this paper uses the AlgaeNet model for optical images and the GANet model for SAR images.

We have made appropriate supplementary explanations in the manuscript, as shown in the following passage: "It is worth noting that although the AlgaeNet model itself also has the ability to detect green algae in SAR images (Gao et al., 2022), compared with the AlgaeNet model, the proposed GANet model has higher green algae extraction accuracy with the help of some improvement strategies, including texture feature enhancement information (Guo et al., 2022)."

13. Why is the image slice size for the AlgaeNet model 128x128 pixels, while for the GANet model it is 256x256 pixels? Please clarify the reasoning behind these different dimensions.

Due to limited computer memory, it is usually necessary to slice satellite images when training AI models. The size of the slice is generally $2^n \times 2^n$ pixels. When the satellite image slice is large, the AI model can more easily learn the overall features on a larger scale. Conversely, if the image slice is small, this overall feature learning ability will be limited to a certain extent, affecting the AI model's 'receptive field' for the image.

The resolution of MODIS images is 500 m, while the resampled SAR images have a resolution of 30 m. For MODIS images, a 128-pixel slice corresponds to a ground area of (500 * 128 / 1000)^2 = 4,096 km². For SAR images, a 256-pixel slice corresponds to a ground area of (256 * 30 / 1000)^2 = 58.98 km². The larger the ground area, the more change information the slice contains. This results in a larger receptive field for the deep-learning model, allowing it to learn more universal information. Through experiments, we found that 128×128 pixels for the AlgaeNet model and 256×256 pixels for the GANet model are very suitable settings. These two AI models can achieve

good feature learning ability and green algae extraction accuracy with these slice sizes.

14. In Table 2, why is the number of testing samples smaller for the new GANet model?

The number of testing samples, 267, is for the retrained GANet model applied to SAR images with a 30m resolution; and the number of testing samples, 2124, is for the GANet model applied to SAR images with a 10m resolution.

15. Is the comparison in Figure 11 meaningful? What is the difference between the two calculations of weekly average coverage?

This is the same question as Question 5, and we have removed previous Figure 11 and Section 3.2.1 from new manuscript.

The first method is to count the number of pixels based on the distribution of green tide coverage provided by the weekly product we released, and then directly calculate the average green tide area for each week. The second method is to first count the area of daily green tide coverage pixels for each day, and then average all daily areas within a week to get the weekly average green tide area. The difference between the two methods is that the former (the weekly product we released) provides both the coverage and distribution of green tides, as well as the green tide area for this coverage, while the latter only provides the overall green tide area for each week (a specific number) without indicating the actual distribution of green algae. To avoid ambiguity, we have removed previous Figure 11 and Section 3.2.1 according to the suggestion from Comment 5.

---

## Author Comment (AC2)

**Response to Reviewer's Comments**

With the help of machine learning, Dr Gao et al., provides a nice dataset of weekly green tide in the Yellow Sea. Several methodology and data sources were considered. They performed a cross validation to show the quality of the generated dataset. I believe this dataset will be useful for the further studies with other purposes; for example, a joint analysis with the wind or sea surface velocity, to have a better understanding the underlying dynamics. I would like to recommend a publication after several minor revisions. I list my comments below.

1 in line 16: There should be a space between ")" and "This".

A space was added between ")" and "This" in Line 16.

2 Figure 2, 9, 11, 12: the quality of these figures are bad. Please provide a high resolution version of these figures.

High resolution versions of these images are provided in new manuscript.

3 line 140: please provide a value of "the submerged portions", for example, 0.5 meter beneath the sea surface.

The depth to which MODIS can detect underwater signals depends on several factors, including water clarity and the specific wavelengths of light being measured. Generally, MODIS can detect signals in clear ocean waters up to a depth of around 10 to 20 meters. In the Yellow Sea, the detection depth of MODIS is generally limited due to high turbidity and a significant amount of suspended sediments. Under such conditions, the effective detection depth is typically much shallower compared to clearer waters. Specifically, in the Yellow Sea, MODIS is usually able to detect underwater signals to a depth of approximately 5 to 10 meters. Ulva prolifera algae rely on the thalli's hollow tubular structure to float on the sea surface, and the submerged part does not exceed 1–2 meters (Ding et al., 2019; Gao et al., 2022). Therefore, MODIS can effectively detect green algae on the water surface and submerged portions up to 1–2 meters beneath the sea surface. Therefore, "the submerged portions" was revised to "the submerged portions of 1-2 meters".

Ding, L., and Luan, R.: The taxonomy, habit, and distribution of a green alga enteromorpha prolifera (ulvales, chlorophyta), Oceanologia et Limnologia Sinica, 40(8), 68–71, 2009.

Gao, L., Li, X., Kong, F., Yu, R., Guo, Y., and Ren, Y.: AlgaeNet: A deep-learning framework to detect floating green algae from optical and SAR imagery, IEEE J. Sel. Top. Appl., 15, 2782-2796, https://doi.org/10.1109/JSTARS.2022.3162387, 2022.

4 line 162: please remove the duplication of "the" when mention "The fifth generation atmospheric reanalysis data"

The redundant "the" has been removed. Additionally, another reviewer believes that the data quality of environmental factors such as HYCOM and EAR5 in the Yellow Sea cannot be guaranteed and suggested removing the relevant part of the environmental factor analysis. We accepted this suggestion, and the related Section 3.5 was removed.

5 line 203: please provide the full name of "VGG16"

The full name of VGG16, Visual Geometry Group 16, is provided in the introduction section when it first appeared.

6 line 206: "We used the unique physical multichannel combination of all bands of MODIS surface reflectance products as input". Is it possible to have an optimization combination of these bands? Or in other way, do we have contamination problem when all bands are involved?

The AI model can automatically assign weights to each input channel during the training process, preventing contamination issues with any physical input bands. Additionally, we can use the AI model to screen multichannel combinations through ablation experiments and determine the optimal combination.

7 line 215: "1.10 km2" should be "1.10 km^2"

"1.10 km2" was revised to "1.10 km$^2$"

8 line 232: "256 256" pixels should be "$256\times 256$"

"256 256" was revised to "256×256".

9 line 233: "These enhancements have ... to 85.41%". Please specific from what value to "85.41%". 我们进行了补充 These enhancements have raised the model's performance to 85.41% (see Table 2)

These enhancements have raised the model's performance to 85.41% **from past 78.58%**

10 Figure 5: "view of the white square part". But in the figure 5, it is a green square. Please correct this typo.

We modified the caption in Figure 5b: "view of the green square part".

11 line 264: "beneath a certain water depth": see comment 3. please provide a value here.

Similarly, the response to comment 3, "beneath a certain water depth" was revised to "beneath a certain water depth of 1-2 meters".

12 Figure 8: the box in the left panel is unclear.

Figure 8 (Figure 10 of the new manuscript) has been revised to make the box in the left panel clear, as shown below.

[Figure]

Figure 10. Green tide detection result analysis from randomly selected optical images

13 line 320: the terminology "dissipation" is used. Is it possible to find another proper terminology? This is because in the fluid dynamics, "dissipation" means something else. For example, the energy dissipation means the conversion the kinetic energy to heat.

The term "dissipation" in the full manuscript was revised to "decaying".

14 Figure 9: For a large value variation, we often use log-log plot. I strongly suggest authors to replot this figure in a log-log view to see possibility of lognormal or other distribution of areas.

Figure 9 (Figure 11 of the new manuscript) illustrates the frequency of occurrence of green algae patches of varying sizes as detected by MODIS. The data reveals that large patches (>100 km²) are less common, while small patches (<100 km²) occur more frequently, suggesting that the green tide in the Yellow Sea predominantly consists of smaller green algae patches. The size of these patches influences the satellite's ability to detect them. Although we attempted to replot the figure using a log-log scale to explore the possibility of a lognormal or other distribution, no clear pattern emerged. Consequently, we redrew the figure with the vertical axis in a log scale to convey the data more accurately.

[Figure]

Figure 11. Statistical analysis of green algae patch size derived from optical imagery.

15 Figure 10: please keep the label of "(g)" in the same style as others.

We revised the labels and subfigure numbers in Figure 10 (Figure 12 of the new manuscript) to maintain a consistent style.

[Figure]

Figure 12. Randomly selected SAR image detection results on the Subei Shoal (a-f) and the entire Yellow Sea (g-j) regions. (Date: June 22, 2018, and June 30, 2021).

16 Figure 11: R^2=1 and RMSE=0 seems too good to be true. Please double check this result.

Figure 11 in previous version shows two methods for calculating the monthly average green tide coverage. Their completely consistent results are expected and confirmed because their data sources are identical, both using daily green tide coverages, and the weekly product is also derived from daily data fusion. The only difference is that one method uses a weekly product derived from our daily data fusion, while the other method directly uses statistical daily data.

The first method is to count the number of pixels based on the distribution of green tide coverage provided by the weekly product we released, and then directly calculate the average green tide area for each week. The second method is to first count the area of daily green tide coverage

pixels for each day, and then average all daily areas within a week to get the weekly average green tide area.

The difference between the two methods is that the former (the weekly product we released) provides both the coverage and distribution of green tides, as well as the green tide area for this coverage, while the latter only provides the overall green tide area for each week (a specific number) without indicating the actual distribution of green algae.

Another reviewer felt that this comparison was not very meaningful. Therefore, to avoid ambiguity and follow the suggestion of another reviewer, we have removed previous Figure 11 and Section 3.2.1 from new manuscript

17 line 370: the Gompertz curve model is used to fit the data without further justification. Please provide more comments here.

The Gompertz curve model is a widely used mathematical model for describing growth processes and has been particularly effective in modeling biological phenomena, such as population growth, tumor growth, and the spread of diseases. Its applicability extends to various domains where growth initially accelerates rapidly but then slows down as it approaches an asymptotic limit. The growth process of Yellow Sea green tide outbreaks also meets these characteristics. The decaying process of the green tide can be seen as the reverse process of the growth stage. Previous studies have demonstrated a good empirical fit to other similar types of data, including green tide data (Winsor, 1932; Xu et al., 2023).

Winsor, C.P. The Gompertz curve as a growth curve. Proc. Natl. Acad. Sci. USA 1932, 18, 1

Xu S, Yu T, Xu J, Pan X, Shao W, Zuo J, Yu Y. Monitoring and Forecasting Green Tide in the Yellow Sea Using Satellite Imagery. Remote Sensing. 2023; 15(8):2196. https://doi.org/10.3390/rs15082196

---

## Author Response (AR2)

**Response to Reviewer's Comments**

The authors have satisfactorily addressed my prior comments and suggestions. I believe this dataset holds value for future research purposes. I have only one additional suggestion regarding Figure 11 in the revised manuscript. Initially, I recommended replotted this figure to explore potential exponential or power laws. The authors created a new version of the figure using the logarithm of the histogram. Viewing Figure 11 (a) on a log-log scale reveals a clear power-law with an experimental scaling exponent of 2.40. I suggest that the authors include this power-law in the revised manuscript. I am ready to recommend this manuscript for publication once this change is made without requiring another round of reviews.

Reply: Thanks for your suggestion! We conducted a power-law analysis on each sub-graph in Figure 11 and confirmed the reviewer's observation: the size of the green tide strip follows a clear and definite power law. Consequently, we redrew Figure 11, added power exponential fitting curves to each sub-graph, and revised the original text as follows:

"Figure 11 also illustrates the frequency of occurrence of green algae patches of varying sizes as detected by MODIS. The data reveals that large patches (>100 km²) are less common, while small patches (<100 km²) occur more frequently, suggesting that the green tide in the Yellow Sea predominantly consists of smaller green algae patches. The size of these patches influences the satellite's ability to detect them. The different sizes and the corresponding counts of the green tide patches also reveal a clear and definite power-law with an experimental scaling exponent, i.e., $y=bx^a$ ($-2.28 < a < -2.97$ and $3 \times 10^5 < b < 5 \times 10^5$). "

[Figure]

**Figure 11. Statistical analysis of green algae patch size derived from optical imagery, with the vertical axis in a log scale.**